# Fungal secondary metabolism is governed by an RNA-binding protein CsdA/RsdA complex

Zili Song[1,2], Shuang Zhou[1], Hongjiao Zhang[1,2], Nancy P. Keller [3], Berl R. Oakley [4], Xiao Liu [1] & Wen-Bing Yin [1,2]

Production of secondary metabolites is controlled by a complicated regulatory network in eukaryotic cells. Several layers of regulators are involved in this process, ranging from pathway-specific regulation, to epigenetic control, to global regulation. Here, we discover that interaction of an RNA-binding protein CsdA with a regulator RsdA coordinates fungal secondary metabolism. Employing a genetic deletion approach and transcriptome analysis as well as metabolomics analysis, we reveal that CsdA and RsdA synergistically regulate fungal secondary metabolism comprehensively. Mechanistically, comprehensive genetic and biochemical studies prove that RsdA and CsdA co-localize in the nucleus and physically interact to achieve their functions. In particular, we demonstrate that CsdA mediates *rsdA* expression by binding specific motif "GUCGGUAU" of its pre-mRNA at a post-transcriptional level. We thus uncover a mechanism in which RNA-binding protein physically interacts with, and controls the expression level of, the RsdA to coordinate fungal secondary metabolism.

Microbial natural products, known as secondary metabolites (SMs), continue to drive novel discoveries in chemistry, biology, and medicine and remain the best sources of drugs and drug leads[1–3]. Many of these compounds serve crucial functions for the microorganism itself and play significant roles in interactions with other organisms. Notable examples include the UV-resistant DHN-melanin[4] and the anti-fungal phenazine synthesized during bacterial-fungal interactions[5]. Many of them have important pharmacological activities and are widely used in health, medicine, and agriculture[6]. Exemplars include the clinically instrumental antibiotic penicillin[7], the immunosuppressant cyclosporine, and the cholesterol-lowering compound lovastatin[8]. However, discovering SMs with new structures using traditional approaches has become increasingly arduous, creating serious challenges in the development of new drugs. Extensive genomic data and bioinformatic analyses indicate that their huge numbers of secondary metabolic biosynthetic gene clusters (BGCs) for SM biosynthesis[9–11]. Only a small fraction of the products of SM biosynthetic genes have been identified because SM BGCs are silent under laboratory conditions[12–14]. To overcome this challenge, the development of innovative approaches is not only imperative but also critical. Gene regulation strategies have been effectively devised and implemented to activate dormant or low-expressed BGCs, enabling the exploration of novel fungal natural products[15]. Nonetheless, the intricate mechanisms underlying the regulation of SM genes in fungi remain only partially elucidated.

The production of SMs is controlled by a complex regulatory network involving multiple layers of regulation including pathway-specific regulation, epigenetic regulation, and global regulation[16–18]. Pathway-specific regulation is often mediated by pathway-specific transcription factors (PSTFs) located inside BGCs that bind to specific DNA sequences in the promoters of the other genes in the BGC. Among these known BGCs, 12 BGCs (42.3%) contain 16 PSTFs in *Aspergillus nidulans*, and 10 BGCs (55.6%) contain 12 PSTFs in *Aspergillus fumigatus*[19]. To illustrate, AflR, a TF, specifically regulates aflatoxin and sterigmatocystin synthesis in *Aspergillus flavus* and *A. nidulans*[20,21], respectively; GliZ regulates gliotoxin synthesis in *A. fumigatus*[22]. In addition, the epigenetic remodeling of the chromosome landscape by

[1]State Key Laboratory of Mycology, Institute of Microbiology, Chinese Academy of Sciences, 100101 Beijing, PR China. [2]Savaid Medical School, University of Chinese Academy of Sciences, 100049 Beijing, PR China. [3]Department of Medical Microbiology and Immunology, University of Wisconsin-Madison, Madison, WI 53706, USA. [4]Department of Molecular Biosciences, University of Kansas, Lawrence, KS 66045, USA. e-mail: yinwb@im.ac.cn

altering transcriptional accessibility to BGCs plays a significant role in regulating the expression of cryptic fungal BGCs[23–25], by modifying proteins such as the histone deacetylase HdaA and the methyltransferase CclA[26,27]. Fungal secondary metabolism is also regulated by global regulators[13]. Examples include the regulation of gibberellin biosynthesis in *Fusarium fujikuroi* by the nitrogen regulator AreA[28], the regulation of patulin biosynthesis in *Penicillium expansum* by the carbon regulator CreA[29], and the regulation of sterigmatocystin biosynthesis in *A. nidulans* by the pH regulator PacC[30]. In fact, fungal SM regulation does not follow a strict hierarchical regime in most cases, multiple levels of regulation interact together[15]. To illustrate, the global regulator LaeA regulates ~50% of BGCs and forms a heterotrimeric velvet complex with VelB and VeA to regulate fungal secondary metabolism and development in response to light signals[31,32]. In epigenetic regulation, the S-adenosyl methionine binding site in LaeA also implies that LaeA might influence the accessibility of binding factors to chromatin regions of BGCs[15,31]. Very recently, we identified a regulator of SM, PfRsdA, in a plant endophytic fungus *Pestalotiopsis fici*. A NCBI BLAST search of PfRsdA indicated the presence of a single predicted PAT1 domain at the *N*-terminus with an unclear function. However, despite the limited domain prediction, PfRsdA is involved in the regulation of over 50% of BGCs[33].

In exploring how RsdA regulates secondary metabolism comprehensively, we have discovered that it is part of an RNA-binding protein (RBP) complex suggesting that it works through a post-transcriptional regulatory mechanism. In this study, we have designated the RBP CsdA because it acts as a coordinator of secondary metabolism and development in fungi. We demonstrate that the loss of *csdA* results in comprehensive changes of fungal secondary metabolism in *P. fici*. Targeted gene deletions in combination with comprehensive transcriptome and metabolomic analysis reveal that RBP CsdA interacts with RsdA comprehensively regulates fungal secondary metabolism. The process further gets confirmed by an in vitro pull-down assay and in vivo bimolecular fluorescence complementation (BiFC) assays. Interestingly, CsdA influenced the expression level of *rsdA* by pre-mRNA splicing at a specific motif GUCGGUAU and furthermore determined the functional mode of RsdA. The post-transcriptional regulatory model of RBP complex broadens our understanding of complex fungal regulatory networks of fungal secondary metabolism and provides a potential approach for the exploitation of bioactive fungal natural products.

## Results

### Identification of CsdA, an RNA-binding protein, associated with the regulator RsdA

Previous studies have demonstrated that RsdA and its homologs comprehensively regulate the secondary metabolism of various fungi such as *A. flavus*[34], *A. nidulans*[35], and the plant endophytic fungus *P. fici*[33]. We presumed that RsdA has a common regulatory mechanism on secondary metabolism in filamentous fungi. In order to investigate the function of RsdA, we drew inspiration from the comprehensive regulatory role of the trimeric VelB/VeA/LaeA complex and the KdmB-EcoA-RpdA-SntB chromatin complex in fungal development and secondary metabolism[25,32], as well as the presence of an unknown functional domain in RsdA. Based on these observations, we hypothesized that RsdA may engage in interactions with other proteins, potentially forming complexes that contribute to its regulatory function. We therefore used a combination approach including pull-down assays[36], liquid chromatography-mass spectrometry (LC-MS), and multi-omics analysis to find its interacting proteins (Fig. 1a). Pull-down of purified bait protein RsdA (PFICI_10532) (Supplementary Fig. 1a) with total soluble proteins from *P. fici* (Fig. 1b and Supplementary Fig. 1b), resulted in the identification of 348 proteins in reaction solutions by Mascot ($p < 0.05$) software (Supplementary Data 1). After removing

duplicate proteins in the RsdA and total protein lanes, a total of 201 proteins were specifically captured by RsdA (Fig. 1b). The mass spectrometry scores of these 201 proteins were then combined with gene expression analysis of previous transcriptome data[33] of an Δ*rsdA* mutant (Fig. 1c). A variety of proteins with high scores or significant changes in gene expression were retrieved. To narrow down target proteins, differential bands (Supplementary Fig. 1b) from a second batch pull-down were cut out and identified by LC-MS. Seven of these 10 proteins that had high scores and were identified repeatedly were selected for further analysis (Fig. 1c and Supplementary Fig. 1c). To avoid missing some of the target genes, we also selected three genes that displayed significant changes (Log$_2$foldchange = −6.21 (*PFICI_13471*), −6.04 (*PFICI_04808*), 5.66 (*PFICI_13986*)) in the transcriptome data of the Δ*rsdA* mutant (Fig. 1c). Ten candidate genes in all were chosen.

We presumed that proteins associated with RsdA function should influence fungal development and secondary metabolism. Each of these 10 candidate genes was consequently deleted individually in *P. fici* employing a previously established CRISPR/Cas9 method[37] (Supplementary Fig. 1d). Growth/phenotype examination and SM analysis of those mutants (Supplementary Fig. 1e–g), revealed that only the *PFICI_10709* deletion mutant exhibited slow growth and chemical profile changes, while other mutants were similar to the control strain (Fig. 1d). This suggests that PFICI_10709 may function in association with RsdA (Fig. 1e, f); thus, we designated PFICI_10709 as CsdA, a coordinator of secondary metabolism and development. Blastp analysis of PFICI_10709 revealed that it contains one RNA recognition motif (RRM) and two Zn-Fingers (ZnFs), indicating that it is an RBP (Supplementary Fig. 1h).

### CsdA and RsdA are ubiquitous in filamentous fungi and comprehensively regulate the development and metabolism

Knowing that RsdA is ubiquitous in fungi[33], we explored the distribution of RsdA or CsdA homologs in 1474 or 1535 species (Supplementary Fig. 2), respectively, by a systematic search of fungal genome data deposited in the NCBI database. Multi-gene combined analyses of *lsu*, *ssu*, *rpb1*, *rpb2*, *its*, and *ef1* from identified 163 species were incorporated to construct a phylogenetic tree[38] (Supplementary Data 2). The results displayed that CsdA and RsdA were widely distributed in Ascomycota, including Sordariomycetes, Leotiomycetes, Dothideomycetes, Eurotiomycetes, and Pezizomycetes. Proteins homologous with CsdA and RsdA showed more than 60%/40% identity and 70%/60% coverage, respectively. In contrast, CsdA and RsdA exhibited low identity/coverage of ca. 30%/20% in Mucoromycota and Basidiomycota (Fig. 2a). These results indicate that CsdA and/or RsdA may have similar functions in Ascomycota, but these may differ in Basidiomycota and Mucormycota.

To confirm the functioning of CsdA and RsdA in the regulation of the development and metabolism of *P. fici*, the phenotypes and metabolomics of the Δ*csdA* and Δ*rsdA* mutants[33] were compared and analyzed in comparison to an isogenic control strain. Similar to the Δ*rsdA* mutant, the growth rate of the Δ*csdA* mutant was dramatically slower over a 5-day period than the control strain (Fig. 2b). Interestingly, stereomicroscopic observation of those two deletion mutants demonstrated that the number of conidiomata increased obviously in the Δ*csdA* mutant but decreased in the Δ*rsdA* mutant in comparison to the control (Fig. 2c). Consistent with these observations, quantitation revealed that the Δ*csdA* strain showed a 60-fold increase in conidial number, while the Δ*rsdA* strain showed a 13-fold decrease compared to the control (Fig. 2d). Deletion of *rsdA* in the Δ*csdA* background can complement the conidial number defect, suggesting that the effects of the two deletions are independent and additive in some cases (Supplementary Fig. 3). Usually, a conidium of *P. fici* contains five specialized cells with three median concolorous cells[4,39]. However, some of the conidia in both deletion

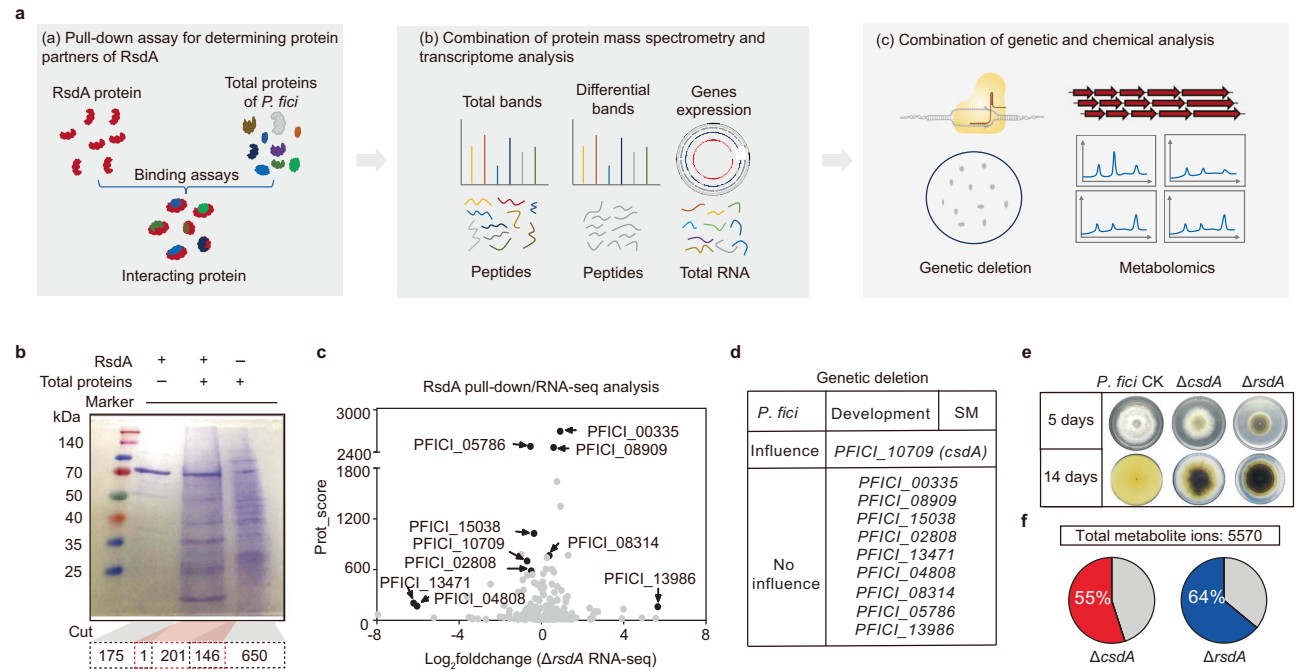

**Fig. 1 | Identification of the RsdA-associated protein CsdA. a** Workflow of RsdA-associated proteins screening. **b** Brilliant blue G-stained 12% SDS-polyacrylamide gel electrophoresis of the RsdA-associated proteins. Lane 1: marker. Lane 2: purified RsdA protein (80.4 kDa), control group. Lane 3: proteins captured by RsdA from the total proteins of *P. fici*, experimental group. Lane 4: total proteins extracted from *P. fici*, control group. The dashed lines below represent the number of proteins identified for lanes 2–4 by LC-MS, respectively. A total of 201 proteins with positive scores and 146 proteins with negative scores were screened in the experimental group after subtracting the mass score of the control group, respectively. **c** Proteins specifically captured by RsdA were scored and analyzed for gene expression. Black spots show high protein scores or significant changes in gene expression. **d** Ten genes targeted for deletion in *P. fici* for analysis of development and secondary metabolism. Table showing genes affecting development and secondary metabolite (SM). **e** Morphological phenotypes, and pigment accumulation in *csdA* and *rsdA* deletants in *P. fici*. *P. fici* CK stands for *P. fici* Cas9. **f** Differential ion products from metabolomic data in *P. fici* and its mutants. Statistical analysis was performed by using *t*-test (two-tailed), and the exact *p* values were shown in Supplementary Data 3 (*n* = 2 biologically independent replicates). Differential metabolic ions: $p < 0.05$, |Log$_2$foldchange| > 1. Source data are provided as a Source Data file.

mutants showed abnormal morphologies such as two or four median concolorous cells in 5-celled conidia (Fig. 2c). Moreover, black pigment accumulation was observed in both Δ*csdA* and Δ*rsdA* strains (Fig. 1e). Therefore, our results demonstrate that both RsdA and CsdA play key roles in growth and conidial formation as well as pigmentation of *P. fici*.

Often the development and metabolism of fungi are closely linked. Comparative metabolome analysis revealed that a total of 5570 ion peaks of metabolites were detected (Supplementary Data 3 and Supplementary Fig. 1i). Among them, 3063 (55%) and 3565 (64%) differential ionic products were produced in the Δ*csdA* and Δ*rsdA* mutants (adjusted $p < 0.05$ and |log$_2$foldchange| > 1), respectively (Fig. 1f). For the differential ionic products of the Δ*csdA* strain, 1615 (29%) showed up-regulation and 1448 (26%) were down-regulated (Fig. 2e). In contrast, among the differential ionic products of the Δ*rsdA* strain, 501 (9%) exhibited up-regulation while 3064 (55%) were down-regulated (Fig. 2e). Further Venn analysis[40] revealed that a total of 2218 (39.8%) metabolic ionic products were co-regulated by CsdA and RsdA, of which 1318 (23.6%) were down-regulated and 225 (4%) were up-regulated (Fig. 2f). These results indicated that both *csdA* and *rsdA* have comprehensive regulatory functions in the metabolism of *P. fici*. Interestingly, some SMs, e.g., DHN-melanin, asperpentyn, and iso-A82775C were significantly regulated in each deletant of *csdA* or *rsdA* (Supplementary Fig. 1g).

### Co-regulation of genes related to fungal development and secondary metabolism by CsdA and RsdA

To obtain a comprehensive understanding of which genes or pathways regulated by CsdA and RsdA are associated with fungal growth, development, and secondary metabolism, we carried out a comparative transcriptome (RNA-seq) analysis of the Δ*csdA*, Δ*rsdA*, and control strains. A total of 17,719 unique transcripts were detected. Differentially expressed genes (DEGs) were analyzed between the mutants and the control when the average reads of the corresponding transcripts differed with an adjusted $p < 0.05$. Notably, 1262 (7%) and 5478 (31%) DEGs exhibited more than two-fold changes in expression levels in the Δ*csdA* and Δ*rsdA* mutants, respectively, compared with the control (Supplementary Fig. 4a, b). Further statistical analyses established that a total of 720 genes were co-regulated by CsdA and RsdA (Supplementary Fig. 4c). We identified the genes involved in the regulation of fungal development in *P. fici* by blastp, and they included *stuA*, *medA*, *flbA*, *fadA*, *vosA*, *velB*, etc (Supplementary Fig. 5a). By transcriptome analysis, we found that *vosA*, a gene associated with fungal asexual development, was significantly down-regulated in both Δ*csdA* and Δ*rsdA* strains (Supplementary Fig. 5b, c). This is consistent with our findings of similar conidial morphologies in the Δ*csdA* and Δ*rsdA* mutants (Fig. 2c).

Focusing on secondary metabolism, the transcriptome data of 76 BGC backbone genes[41,42] distributed in 16 scaffolds of the *P. fici* genome were analyzed in depth (Fig. 2g). Among them, the expression levels of 45 (59%) and 54 (71%) of the backbone genes showed significant changes in the Δ*csdA* and Δ*rsdA* mutants, respectively (Supplementary Data 4 and Supplementary Fig. 6). By comparing the differentially expressed backbone genes between Δ*csdA* and Δ*rsdA* mutants, we found that a total of 38 genes (50%) were significantly regulated in both mutants, 24 showed the same expression trend, and 14 showed the opposite expression trend

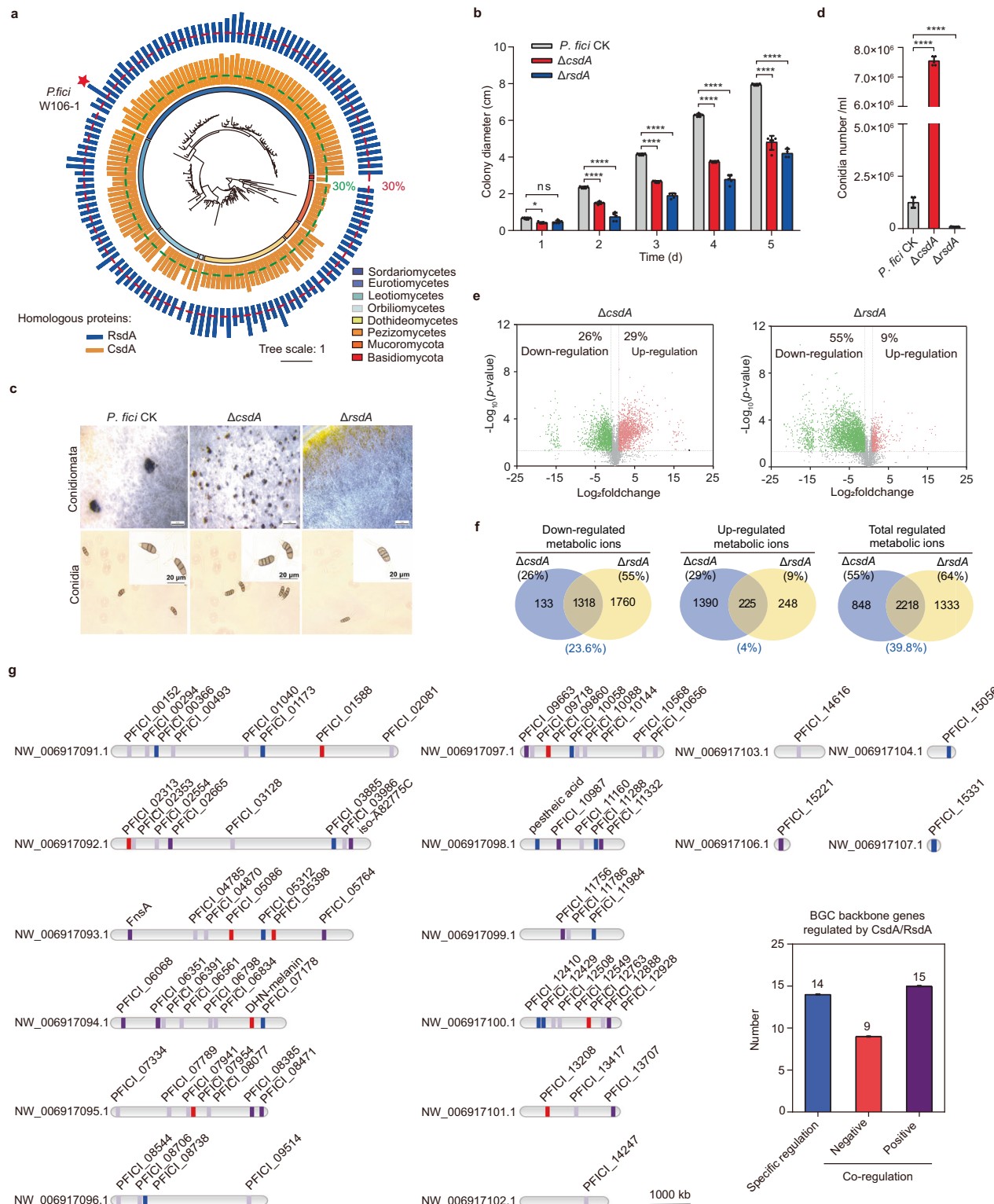

(Fig. 2g). Among the 24 genes with the same expression trend, 9 genes encoding 3 polyketide synthases (PKSs), 1 non-ribosomal polypeptide synthases (NRPSs), 1 terpene synthase (TC) and others were up-regulated, and 15 genes encoding 7 PKSs, 4 NRPSs, 1 TC, 1 NRPS/PKS hybrid enzyme (the FnsA we recently identified[43]) and 2 others were down-regulated in both mutants (Fig. 2g). Taken together, CsdA and RsdA alone or together, regulate fungal metabolic processes by affecting the expression levels of genes related to metabolic processes.

## CsdA and RsdA co-localize in the nucleus and physically interact each other to form RBP complex

To contribute to the understanding of the function of CsdA and RsdA, we initially analyzed the subcellular localization of RsdA and CsdA in *P. fici*. There are two nuclear localization signals in the *C*-terminus of RsdA, DIPDAKKRKFI (533–543 aa) and RVGKRRWSERM (710–720 aa), but not in CsdA (Supplementary Fig. 7a, b). Individually, green fluorescent protein (sfGFP) fused with RsdA and red fluorescent protein (mCherry) fused with CsdA were expressed in *P. fici*. In the

**Fig. 2 | CsdA and RsdA work together to control fungal development and comprehensive metabolism. a** Phylogenetic tree of fungal species that contain both CsdA and RsdA. Star: fungal species shown in this study. Outer columns: homology between RsdA and its homologs. Inner columns: homology between CsdA and its homologs. **b** Measurement of radial growth in the ΔcsdA and ΔrsdA mutants of *P. fici*. **c** The conidiomata and conidia in *P. fici* and their mutants were analyzed microscopically. Top row: scale bar = 2 mm, bottom row: scale bar = 20 μm. **d** Conidial production in *P. fici* and their mutants. **e** Volcano plots show the differentially regulated metabolic ion products of *P. fici* mutants versus the control. **f** Metabolomic analysis of total metabolic ions co-regulated by CsdA and RsdA in *P. fici*. Blue: the minimum percentage of metabolites regulated by CsdA and RsdA.

**g** The distribution and expression levels of secondary metabolite biosynthesis gene clusters were analyzed by transcriptome data in *P. fici* and its mutant. The 76 BGCs were distributed in 16 scaffolds of the *P. fici* genome. Gray: backbone genes not regulated by *csdA* and *rsdA*. Blue: backbone genes regulated solely by *csdA* or *rsdA*. Red: backbone genes negatively regulated by *csdA* and *rsdA*. Purple: backbone gene positively regulated by *csdA* and *rsdA*. Specific regulation: backbone genes are significantly regulated by *csdA* and *rsdA*, but with opposite patterns of regulation. Data are presented as means ± SD (*n* = 5 biologically independent replicates) (**b, d**). Statistical analysis was performed by using two-way ANOVA (ns not significant, *p* = 0.0863. Significant at *$p$ = 0.0341, ****$p$ < 0.0001). Source data are provided as a Source Data file.

recombinant RsdA-sfGFP or CsdA-mCherry strain, green or red fluorescence signals were detected effectively by fluorescence microscopy (Fig. 3a). Furthermore, overlaying the signals with DAPI (4′,6-diamidino-2-phenylindole, a DNA binding dye) fluorescence identified that RsdA and CsdA were both localized in the nucleus (Fig. 3a). Based on the above results, an in vitro His pull-down was carried out to confirm the interaction between the two proteins (Fig. 3b). RsdA immobilized on nickel magnetic beads was co-incubated with the purified target protein GST-CsdA (Supplementary Fig. 10b), and the target bands were detected by western blotting (Fig. 3b). In the input samples, RsdA-His (80.4 kDa) and GST-CsdA (93.6 kDa) were distinctively detected by anti-His and anti-GST antibodies, respectively, signifying the feasibility of a pull-down assay for sample detection. In the experimental group, we detected that GST-CsdA was efficiently pulled down by RsdA-His (Fig. 3b), confirming direct physical interactions between RsdA and CsdA. To further validate the interaction between RsdA and CsdA in the nucleus, a BiFC assay was conducted in *P. fici* (Fig. 3c). The recombined yellow fluorescent protein (YFP) signals from RsdA-YFP^[N-terminal] and CsdA-YFP^[C-terminal] in the fusant exhibited co-localization with the nuclear DAPI, while no YFP fluorescence signals were detected in the control (Fig. 3c). These results indicate that the physical interaction between CsdA and RsdA promoted the physical interaction of the two portions of YFP in the nucleus.

To determine the amino acid sequence on which CsdA interacts with RsdA, AlphaFold2[44] was used to predict the three-dimensional structure of CsdA and RsdA as well as their dimer structure (Supplementary Fig. 7c, d). The three-dimensional structure scores of CsdA and RsdA were 71.64 and 60.98 (Supplementary Figs. 8 and 9), respectively. Therefore, we predicted the dimer model of CsdA and RsdA to find the interaction sites between CsdA and RsdA (Supplementary Fig. 10a). The dimer structure model showed that the potential interaction sites of RsdA were S668, R669 and other sites in the *C*-terminal region, and the potential interaction sites of CsdA were in the exonuclease domain in the *N*-terminal region (Fig. 3d and Supplementary Fig. 10a). Next, we expressed and purified GST-CsdA^ΔN-terminal, GST-CsdA^ΔRRM, GST-CsdA^ΔZnF, and RsdA^ΔC-terminal -His proteins in vitro (Supplementary Fig. 10b). By pull-down assays of the GST-CsdA^ΔN-terminal, GST-CsdA^ΔRRM or GST-CsdA^ΔZnF with RsdA as well as GST-CsdA with RsdA^ΔC-terminal in vitro, we only detected weak bands in the reaction of GST-CsdA^ΔRRM or GST-CsdA^ΔZnF with RsdA-His protein (Fig. 3e). In addition, we deleted the *N*-terminus of CsdA or the *C*-terminus of RsdA in *P. fici* to verify the function of the CsdA/RsdA complex. The results showed that both CsdA^ΔN-terminal and RsdA^ΔC-terminal mutants showed slow growth, abnormal conidial morphology, and different metabolic profiles compared with the control strain (*P. fici* CK). This suggests that CsdA regulates the growth, development, and metabolism of fungi by interacting with RsdA (Fig. 4). This result was also confirmed by the mutation of amino acid residues (H26-Y29) which were in the predicted CsdA and RsdA interaction sites (Fig. 4). To further address the function of CsdA, RsdA and their complex in metabolic regulation, we performed functional enrichment and metabolic analysis from transcriptomic and metabolomic levels. From the transcriptomic level, we reveal that the functions of CsdA are

involved in carbon metabolism and TCA cycle, RsdA are involved in carbon metabolism and amino acid metabolism, and CsdA/RsdA complex are involved in carbon metabolism and terpenoid biosynthesis. From the metabolomic level, the metabolic ion peaks of 2830 (39%) and 3923 (55%) were significantly changed in the mutation of the binding sites H26-Y29 in the CsdA/RsdA complex or complete deletion of the interaction region (*N*-terminus of CsdA), respectively. These results suggested that this complex has a complicated function in metabolic regulation (Supplementary Fig. 11).

## CsdA mediates *rsdA* expression at the post-transcriptional level via pre-mRNA splicing

Having demonstrated that CsdA and RsdA interact and regulate secondary metabolism in fungi, we aimed to further decipher the function of CsdA as an RNA-binding protein. Therefore, we determined the relative expression levels of *rsdA* and *csdA* by a quantitative real-time PCR (qRT-PCR) in *P. fici* and its mutants, respectively. The expression of *rsdA* was found to be significantly down-regulated in the ΔcsdA mutant, while the expression of *csdA* was not dramatically changed in *rsdA* was deleted (Fig. 5a). These results indicated that CsdA positively regulates the expression of *rsdA*. CsdA is an RBP, and it is generally believed that RBPs are involved in eukaryotic RNA metabolic processes, including alternative splicing, transport, and degradation, at the post-transcriptional level[45,46]. To probe the function of CsdA, the three-dimensional structure of CsdA was further analyzed. The results showed that the RRM domain (PfRRM) and ZnF domain (PfZnF) in CsdA (Supplementary Fig. 8) were similar to the general topology reported for RBPs[47]. By multiple alignments of RRM and ZnF from *P. fici*, *A. fumigatus*, and *A. nidulans* with sequences reported in humans, we identified conserved ribonucleoprotein domains 1 and 2 (RNP1, RNP2) in the fungal RRM domain and a conserved motif (W-X-C-X_{2-4}-C-X_3-N-X_6-C-X_2-C) belonging to the "CCCC" type of ZnF[48] (Supplementary Fig. 12a–d). This suggests a potential function of CsdA in regulating RNA metabolism.

With the comparative evaluation of the changes in abundance of transcripts between the ΔcsdA mutant with the corresponding control strain, we discovered that the expression abundance of *rsdA* in the ΔcsdA mutant was significantly reduced (Fig. 5b). However, the expression abundance of *csdA* in the ΔrsdA mutant did not change significantly (Fig. 5c). This illustrates that CsdA positively regulates the accumulation of *rsdA* in transcripts which was consistent with the above qRT-PCR analysis. To determine whether the expression of *rsdA* was regulated by CsdA at the post-transcriptional level, we examined the mRNA splicing efficiency of *rsdA* in the ΔcsdA mutants of *P. fici* by qRT-PCR[49]. Interestingly, we discovered that the splicing ratio of *rsdA* intron 2 in the ΔcsdA mutant was 1.8-fold higher than that in the control strain, while no significant changes were observed in other introns (Fig. 5d). Moreover, the abundance of intron 2 in the ΔcsdA mutant was measured at the transcriptional (pre-mRNA) and post-transcriptional levels (mRNA), respectively. We found that the abundance of *rsdA* intron 2 in the ΔcsdA mutant was not significantly different at the transcriptional level, but was significantly decreased at the post-transcriptional level by more than 2 folds compared with the control,

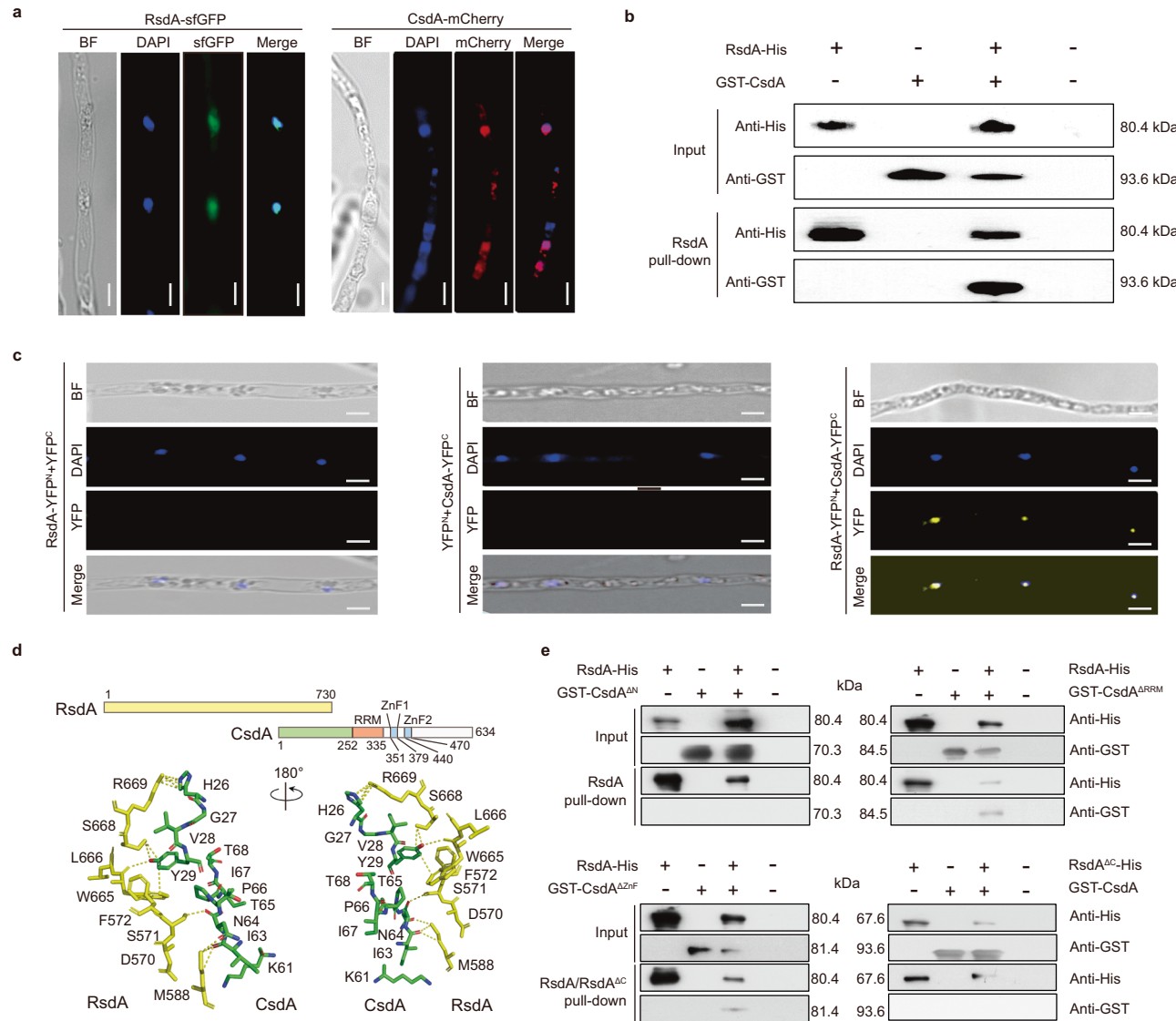

**Fig. 3 | Both CsdA and RsdA are localized in the nucleus and physically interact with each other. a** Subcellular localization of RsdA (left panel) and CsdA (right panel) in *P. fici*. RsdA-sfGFP or CsdA-mCherry localizes in the nucleus and co-localize with DAPI signals. Nuclei in hyphae were stained with 4', 6-diamidino-2-phenylindole (DAPI). BF bright-field. Scale bars, 5 μm. **b** The interaction between CsdA and RsdA was studied by in vitro pull-down assay. **c** The interaction between RsdA with CsdA was studied by in vivo bimolecular fluorescence complementation (BiFC). The strains harboring a single construct (RsdA-YFP^N or CsdA-YFP^C) was used as a negative control. Scale bars, 5 μm. **d** The interaction model of CsdA and RsdA was predicted by Alphafold2. The docking sites on the RsdA amino acid sequence are in the *C*-terminal region and the docking sites of the CsdA amino acid sequence are in the *N*-terminal region. **e** The key sites of CsdA and RsdA interaction were confirmed by in vitro pull-down assays. The *N*-terminal region of CsdA and the *C*-terminal region of RsdA are necessary for their interaction. Source data are provided as a Source Data file.

indicating that the regulatory effect of CsdA is likely post-transcriptional rather than transcriptional levels (Supplementary Fig. 13). Taken together, the RBP CsdA characteristically facilitates gene expression of *rsdA* at the post-transcriptional level by influencing the alternative splicing of pre-mRNA.

## CsdA binds to the specific sequence of *rsdA* pre-mRNA

The RRM or ZnF domain is known as an important functional unit of RBPs. We suspected that CsdA could directly act on the target pre-mRNA through the RRM or ZnF domain. Therefore, we predicted the pre-mRNA structure near the splice site of the target *rsdA* by 3DRNA software (Supplementary Fig. 14) and performed molecular docking with the CsdA protein via the HADDOCK website (Supplementary Fig. 12e). The docking results of CsdA protein and *rsdA* pre-mRNA indicated that sites R368 and R369 in the ZnF domain at the *C*-terminal of CsdA were responsible for recognizing GUCGGUAU on the *rsdA* pre-

mRNA (Supplementary Fig. 12e). The binding sites in the ZnF domain of CsdA are consistent with those in humans, suggesting the conserved function of ZnF domain in eukaryotes (Fig. 5e). Next, we mutated amino acid residues R368-F373 at the binding site, and found that the growth, development, and secondary metabolism of the mutant were consistent with those of the Δ*csdA* mutant (Fig. 4). In addition, the splicing rate of *rsdA* pre-mRNA in the CsdA^R368A-F373A mutant was also increased by 1.6 times, which was consistent with the Δ*csdA* mutant (Fig. 5d). To verify the interaction of CsdA with the target pre-mRNA, we carried out electrophoretic mobility shift assays (EMSAs) using purified GST-CsdA protein and the 5' end GUCGGUAU of intron 2 synthesized in vitro. The 5' end RNAs of intron 1 (GCAGGUCA) and intron 3 (GCCCGUAC) were used as negative controls. As expected, retardation band of intron 2 (GUCGGUAU) migration was observed in the GST-CsdA protein (Fig. 5f). In order to determine the binding activity of CsdA to the target sequence, Isothermal Titration

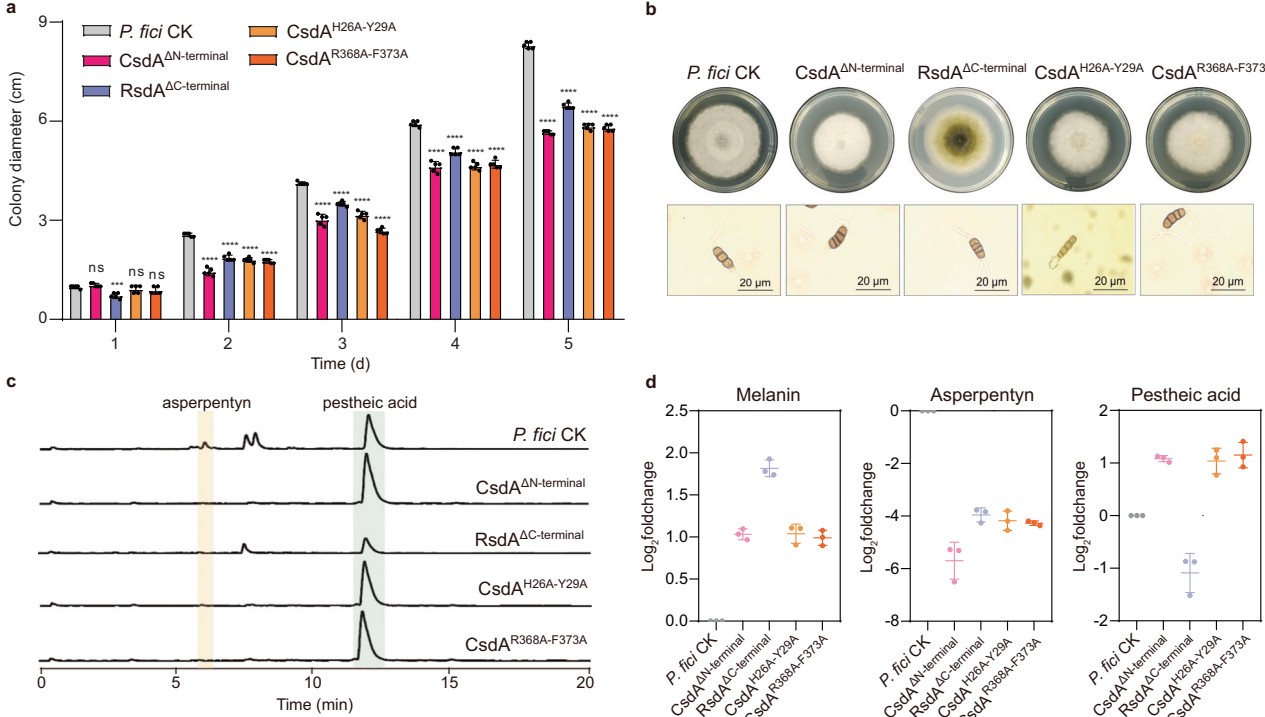

**Fig. 4 | Mutations in the binding sites of CsdA and RsdA affect the development and secondary metabolism of *P. fici*. a** Measurement of radial growth in *P. fici* and its mutant strains ($n$ = 5 biologically independent samples). **b** Comparative analysis of colonies and conidial morphology between *P. fici* and its mutant strains. **c** HPLC analysis of SMs in *P. fici* and its mutant strains. UV absorptions at 254 nm are illustrated. **d** Production of representative SMs in *P. fici* and its mutant strains ($n$ = 3 biologically independent samples). Data are presented as means ± SD (**a**, **d**). Statistical analysis was performed by using two-way ANOVA (ns not significant; CsdA$^{\Delta N\text{-}}$ $^{terminal}$ $p$ = 0.7264; CsdA$^{H26A\text{-}Y29A}$ $p$ = 0.7264; CsdA$^{R368A\text{-}F373A}$ $p$ = 0.3008. Significant at ***$p$ = 0.0002, ****$p$ < 0.0001). Source data are provided as a Source Data file.

Calorimetry (ITC) was used to fit the isotherm. The results show that the $K_d$ value of CsdA and 1×GUCGGUAU or 4×GUCGGUAU is 19.74 ± 3.67 μM or 10.48 ± 5.12 μM (Fig. 5g and Supplementary Fig. 15), respectively. This indicated that the affinity between CsdA and 4×GUCGGUAU was significantly higher than that of 1×GUCGGUAU, further confirming the target sequence of CsdA binding.

## CsdA and RsdA co-regulate the production of representative secondary metabolites

To further understand the mechanisms underlying the comprehensive regulation of secondary metabolism by *csdA* and *rsdA* in filamentous fungi, we precisely analyzed expression of representative co-regulated SMs. DHN-melanin of filamentous fungi is synthesized by a PKS and plays a key role in the protection against UV radiations and in spore cell wall synthesis[4,39]. Therefore, we quantified melanin in the Δ*rsdA* and Δ*csdA* mutants and found that it was up-regulated 5.2-fold and 2-fold, respectively, in comparison with the control (Fig. 6a). Based on our previous study[39], the synthesis of melanin in *P. fici* involves five genes (*pfma*D-H) within and two genes (*pfma*I-J) outside the DHN-melanin gene cluster (*pfma*). By analyzing the expression levels of these seven genes (*pfma*D-J), we found that they all were significantly negatively regulated by CsdA and RsdA, even though the extents of their regulation were different (Fig. 6a). The results were further supported by qRT-PCR (Supplementary Fig. 16a). Evaluating the alterations in intron abundance of *pfma* cluster genes in the Δ*csdA* mutant, no significant changes were found in comparison to the control (Fig. 6b). This suggested that CsdA may regulates DHN-melanin synthesis through RsdA. Analysis of metabolomic data revealed that natural product iso-A82775C, the precursor of the antitumor compound chloropupukeananin[50], was down-regulated more than 8-fold in the Δ*csdA* and Δ*rsdA* mutants, respectively (Fig. 6c). Next, the gene expression levels of the *iac* cluster involved in iso-A82775C[50,51]

synthesis were analyzed from transcriptome data in the Δ*csdA* and Δ*rsdA* mutants. Six genes (*iac*C, E, F, G, H, I) were down-regulated, but the regulatory intensity of *rsdA* was significantly higher than that of *csdA* (Fig. 6c). The results were further supported by qRT-PCR (Supplementary Fig. 16b). In addition, intron abundance of six *iac* cluster genes from the transcriptome data of the Δ*csdA* mutant showed no significant changes indicating CsdA regulates iso-A82775C synthesis through RsdA too (Fig. 6d). In conclusion, the CsdA controls BGCs gene expression levels in filamentous fungi, by specifically regulating alternative splicing of *rsdA*, leading to comprehensive changes in SMs.

## Discussion

The post-genomic era has revealed a large number of SM biosynthetic gene clusters in fungal genomes, in vast excess over the number of compounds that have been identified[6,14]. This can be explained by the fact that most SM BGCs are silent under normal laboratory conditions. SM BGCs are controlled by a complicated gene regulatory network, and the understanding of gene regulation of fungal secondary metabolism is still limited[16]. Compared to other types of regulation, global regulation provides an additional, higher level of regulation for fungal SMs[16]. In order to explore the regulatory mechanism(s) of the RsdA, we conducted a comprehensive study via morphological observations, targeted gene deletions, transcriptomics, metabolomics analysis, and in vitro biochemical studies. Pull-downs of RsdA with total soluble proteins from *P. fici* remarkably revealed an RBP, CsdA that regulates fungal secondary metabolism (Figs. 1 and 2). The unexpected revelation of this RBP's involvement in the comprehensive regulation of fungal secondary metabolism represents a significant discovery. RBPs are important proteins in cells and are involved in multiple post-transcriptional regulatory processes, such as pre-mRNA splicing, transport, turnover, and intracellular localization[45]. Many RBPs have been reported to be involved in the regulation of various human

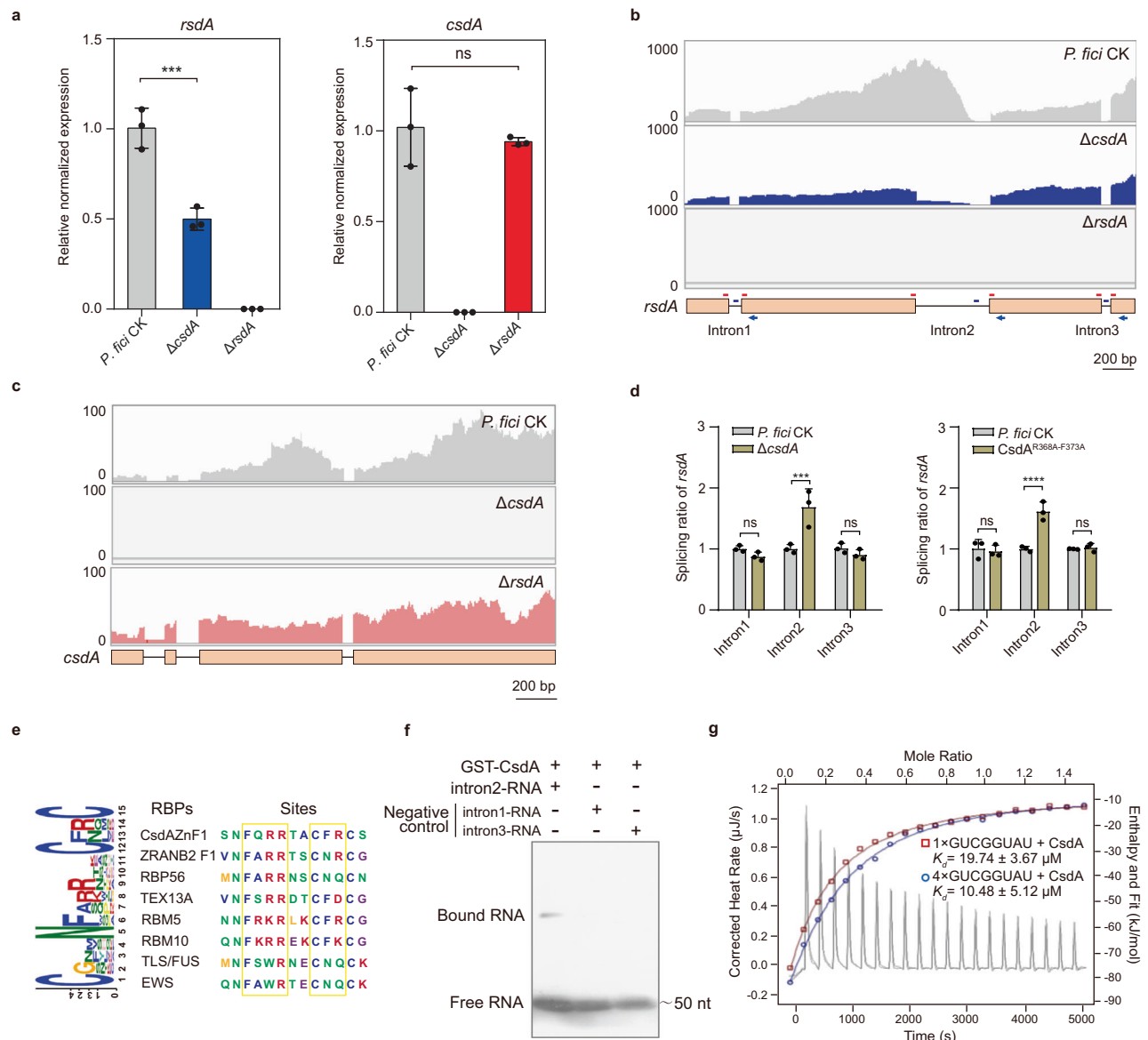

**Fig. 5 | CsdA specifically regulates *rsdA* expression by binding to the motif GUCGGUAU for splicing of pre-mRNA. a** The expression levels of *rsdA* or *csdA* in *P. fici* and its mutants were determined by qRT-PCR. CsdA positively regulates the expression of *rsdA* in *P. fici*. Statistical analysis was performed by using one-way ANOVA (ns not significant, $p = 0.6738$; Significant at ***$p = 0.0003$). **b** RNA-seq analysis of a Δ*csdA* mutant revealed expression abundance of *rsdA* compared to the control strain. The significant change in the 5' end of *rsdA* intron 2 in Δ*csdA* and control strains suggests an alternative splicing role for CsdA. A schematic of the genes is shown at the bottom. The arrows and dashes indicate the positions of the primers used to measure intron splicing efficiency. The red dashes in the diagram represent the primers designed to amplify regions composed of exon-exon junctions after intron removal (splicing), while the blue dashes represent the primers designed within the introns that remain unspliced. **c** RNA-seq analysis from the

Δ*rsdA* mutant revealed the expression abundance of *csdA* compared to the control strain. Values on the *y*-axis indicate reads per 10 million (**b, c**). **d** The splicing efficiency of *rsdA* pre-mRNA in the Δ*csdA* and CsdA[R368A-F373A] mutants were determined by qRT-PCR. The splicing efficiencies were calculated as spliced RNA normalized to the level of unspliced RNA. Statistical analysis was performed by using two-way ANOVA (Left panel: intron1 $p = 0.6356$, intron3 $p = 0.7377$; Right panel: intron1 $p = 0.9351$, intron3 $p = 0.9894$; Significant at ***$p = 0.0002$, ****$p < 0.0001$). **e** Identification of binding motifs in CsdA using multiple EM motif elicitation (MEME) programs. **f** The binding of CsdA specificity to *rsdA* intron 2 was determined by EMSA assay. **g** Isothermal titration calorimeter (ITC) assays showed the binding of CsdA with $1 \times$ GUCGGUAU or $4 \times$ GUCGGUAU. Data are presented as means ± SD ($n = 3$ biologically independent replicates) (**a, d**). Source data are provided as a Source Data file.

diseases as the cofactors of the spliceosome[52,53], such as neurodegenerative diseases, autoimmunity, and cancer[46,54–56]. In *F. graminearum*, the RBP FgRbp1 interacts with U2AF23 causing 47% of the gene transcripts to be spliced[49]. In *Magnaporthe oryzae*, the interaction between RBP35 and CFI25 mediates 3' end alternative splicing of pre-mRNA to regulate fungal development and virulence[57]. RBPs usually consist of multiple domains, such as RRM and ZnF domains. By bioinformatic analysis, CsdA contains one RRM and two ZnF domains (Supplementary Fig. 1h), suggesting that CsdA may be involved in RNA metabolism.

Further analyzing the changes of gene transcripts in the Δ*csdA* mutant revealed that CsdA plays an important role in alternative splicing (Supplementary Fig. 17). Subsequent EMSA and ITC assays demonstrated that CsdA binds to GUCGGUAU on *rsdA* pre-mRNA for alternative splicing to influence gene expression (Fig. 5).

A deletion mutant of CsdA in *P. fici* demonstrated comprehensive changes in secondary metabolism and growth rates (Fig. 2 and Supplementary Fig. 1). These results are similar to those of Δ*rsdA*, CsdA[ΔN-terminal], and RsdA[ΔC-terminal] mutants, suggesting a general

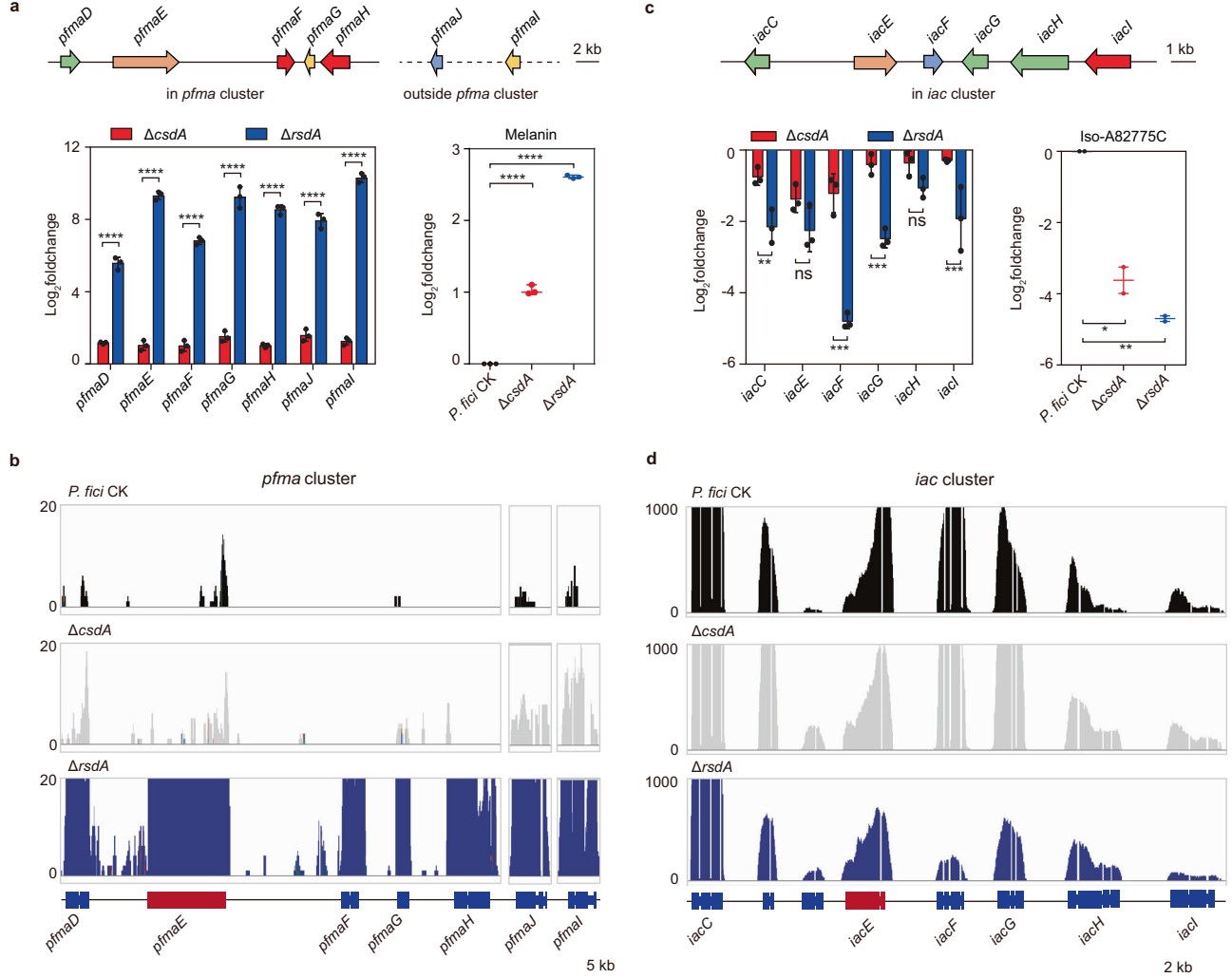

**Fig. 6 | Representative secondary metabolites are co-regulated by CsdA and RsdA.** **a** *pfma* cluster gene expression and melanin production are regulated by RsdA and CsdA in *P. fici*. The *pfma* D-J genes are required for DHN-melanin synthesis. **b** Alternative splicing analysis of genes within *pfma* cluster associated with DHN-melanin synthesis in *P. fici* and its mutants by transcriptome data. **c** *iac* cluster gene expression and terpene iso-A82775C production are regulated by CsdA and RsdA in *P. fici*. The *iac* C, E, F, G, H, I genes are essential for iso-A82775C synthesis. **d** Alternative splicing analysis of genes within *iac* cluster associated with

iso-A82775C synthesis in *P. fici* and its mutants by transcriptome data. The *y* axis represents the abundance of gene expression by transcriptome data (**b, d**). There was no significant change in pre-mRNA splicing between the mutants and the control. Data are presented as means ± SD (*n* = 3 biologically independent replicates) (**a, c**). Statistical analysis was performed by using two-way ANOVA (ns not significant, *iacE p* = 0.1288, *iacH p* = 0.311). Significant at *\*p* = 0.0135, *\*\*p* = 0.0036 (*iacC*) or 0.008 (Δ*rsdA*), *\*\*\*p* = 0.0007, *\*\*\*\*p* < 0.0001. Source data are provided as a Source Data file.

regulatory role of CsdA/RsdA interactions in SM and development. Furthermore, based on in vivo BiFC and in vitro pull-down protein interaction assays, we proved that CsdA and RsdA co-localized in the nucleus, and that the *C*-terminal of RsdA and the *N*-terminal of CsdA are necessary for binding of the two proteins to each other (Fig. 3). The detailed function of CsdA/RsdA complex was determined by the subsequent transcriptomic and metabolomic analysis in their mutants. Transcriptomic data of the Δ*rsdA* or Δ*csdA* mutants in *P. fici* exhibited that 71% or 59% BGC backbone genes were regulated, respectively (Supplementary Data 4). Among them, ~32% of the BGC backbone genes were co-regulated by both CsdA and RsdA (Fig. 2g). Metabolomics analysis of *rsdA* and *csdA* mutants revealed that 64%, 55% of *P. fici* metabolic ion peaks were regulated, respectively (Fig. 1f). Interestingly, 39% and 55% of the metabolic ion peaks were regulated significantly in CsdA^H26A-Y29A and CsdA^ΔN-terminal mutants which the interaction site or region between CsdA and RsdA were disrupted, indicating CsdA/RsdA complex's role in metabolic regulation (Supplementary Fig. 11). Taken together, although we characterized the

functional mechanisms of CsdA and RsdA by identifying their interaction sites and pre-mRNA splicing sites, the precise regulatory mechanism of CsdA/RsdA complex remains complex. Further study of the crystal structure of CsdA/RsdA is helpful to understand its mechanism.

Above all, we propose a model to show how RsdA interacts with CsdA to regulate fungal secondary metabolism (Fig. 8). In this model, CsdA does not directly perform alternative splicing in the regulation of most secondary metabolites in filamentous fungi (Fig. 6). It controls the accumulation of *rsdA* mRNA by alternative splicing to maintain homeostasis during fungal normal growth and metabolism. CsdA homologs show high degrees of amino acid identity in Ascomycota (more than 40%), but lower identity in other eukaryotes due to the relatively small proportion of functional domains in CsdA (Supplementary Fig. 1h), e.g., 24%/46% identity and 22%/6% coverage in plants (*Oryza sativa*)/animals (*Homo sapiens*), respectively. Further phylogenetic analysis of the ZnF domain in CsdA and eukaryotic ZnF domain revealed that the residues around the active sites were conserved

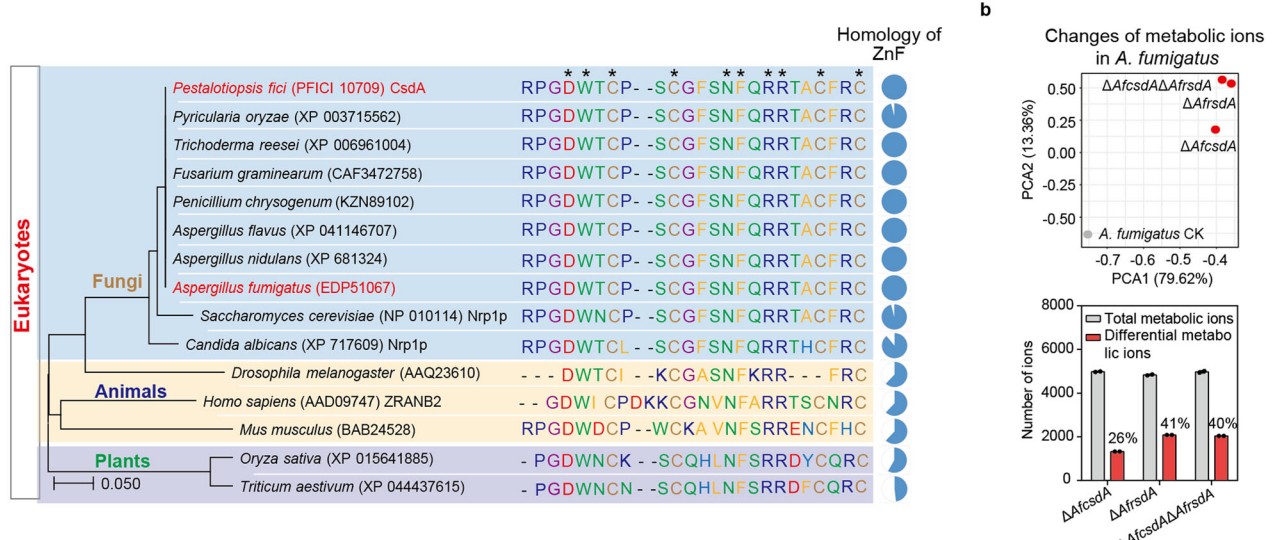

**Fig. 7 | The functional consistency of RNA-binding protein CsdA/RsdA complex in fungal metabolism. a** The phylogenetic tree was constructed based on the amino acid sequences of ZnF from 15 eukaryotes. The conserved binding sites (asterisk) of ZnF in eukaryotes are conserved. **b** The functional consistency of RNA-binding protein CsdA/RsdA complex in fungal metabolism was verified in human pathogenic fungus *Aspergillus fumigatus*. Comparative metabolomics of WT and *csdA* and *rsdA* knockout strains was used to determine the metabolic changes in *A. fumigatus*. Statistical analysis was performed by using *t*-test (two-tailed), and the exact *p* values were shown in Supplementary Data 5 (*n* = 2 biologically independent replicates). Source data are provided as a Source Data file.

(Fig. 7a). To verify that the regulatory mechanism of the RBP CsdA/RsdA complex is prevalent in fungi, we determined and analyzed the metabolome of Δ*Afcsda*, Δ*Afrsda*, and Δ*Afcsda*Δ*Afrsda* mutants in *A. fumigatus*. The results confirmed the comprehensive regulation of CsdA/RsdA complex on fungal metabolism (Fig. 7b and Supplementary Data 5). To further understand the significance of the CsdA and *rsdA* pre-mRNA interaction, we also mutated the binding sites (R368-F373) on CsdA of *P. fici*. The CsdA^R368A-F373A mutant showed similar splicing functionality, development phenotype, and metabolic profile to the Δ*csdA* mutant (Figs. 4 and 5d). This suggests that CsdA regulates fungal development and metabolism by controlling alternative splicing of *rsdA*. Thus, we hypothesized that in the metabolic balance of the fungus when the abundance of *rsdA* expression is reduced, CsdA may promote *rsdA* expression by regulating the stability of intron 2 splicing, thereby producing normal levels of RsdA protein that interacts with CsdA to regulate fungal secondary metabolism (Fig. 8). It also has another possibility that *rsdA* mature mRNA has a shorter half-life than pre-mRNA resulting in lower *rsdA* expression in the Δ*csdA* mutant. A lot of the data are consistent with CsdA simply regulating the amounts of functional RsdA. However, in a few instances, the deletion of *csdA* and *rsdA* gives opposite results. This implies that CsdA functions by additional mechanisms. Further multiple sequence alignment analysis of RRM or the ZnF domain in CsdA and human-derived domains suggesting that the post-transcriptional regulatory mode of CsdA could occur universally in eukaryotic cells (Supplementary Fig. 12).

In this study, we identified a fungal RBP CsdA that coordinates fungal secondary metabolism. We demonstrated that CsdA controls pre-mRNA splicing by binding to the *cis*-element GUCGGUAU on *rsdA*, thereby affecting the expression of RsdA (Fig. 5 and Supplementary Fig. 13). Meanwhile, CsdA physically interacts with RsdA forming a complex to regulate fungal SMs and affect fungal growth (Fig. 8). The ubiquity of this complex in regulating secondary metabolism and growth of filamentous fungi was demonstrated by phylogenetic analysis (Fig. 2 and Supplementary Fig. 1). This study reveals a transcriptional and post-transcriptional crossed mode of secondary metabolic regulation mediated by RBP CsdA. It broadens our understanding of the complex regulatory networks of fungal secondary metabolism and provides insight into the regulatory patterns of RBP complexes across eukaryotes.

## Methods
### Strains, media, and culture conditions
The strains used in this study are listed in Supplementary Data 6. *Pestalotiopsis fici* CGMCC3.15140 (WT) and its transformants were grown at 25 °C on potato dextrose agar (PDA) or in potato dextrose broth (PDB) with appropriate antibiotics as required[33]. *Aspergillus fumigatus* CEA17.1, CEA17.2 and their transformants were grown at 37 °C on glucose minimum medium (GMM) with appropriate supplements corresponding to the auxotrophic marker or antibiotics[11,58]. *A. fumigatus* and its transformants were cultivated in liquid GMM medium supplemented with 0.5% (w/v) yeast extract at 25 °C for 5 days to extract secondary metabolites (SMs) and for metabolome analysis. *Escherichia coli* DH5α and BL21 were used for plasmid construction and protein expression in LB medium (1% tryptone, 0.5% yeast extract, 1% NaCl) with appropriate antibiotics, respectively. *Saccharomyces cerevisiae* BJ5464-NpgA[59] and its transformants were grown at 30 °C on synthetic dextrose complete (SDC) medium with appropriate supplements corresponding to the auxotrophic markers[42] to construct green fluorescent protein (sfGFP) or red fluorescent protein (mCherry) expression vector.

### Gene cloning and plasmid construction
The plasmids and oligonucleotides (Sangon Biotech Co. Ltd, China) used in this study are listed in Supplementary Data 7 and 8, respectively. To construct plasmids and linear DNA fragments, high fidelity polymerase, TransStart ® FastPfu DNA polymerase (Trans-Gen Biotech, China) and Q5 (New England Biolabs) were used. Diagnostic PCRs were performed with 2 × T5 Super PCR Mix (Tsingke Biotechnology Co., Ltd, China). PCR amplifications were performed on a C1000™ Thermal Cycler from Bio-Rad (Hercules, CA) according to standard manufacturer's instruction of different polymerases.

In order to obtain deletion mutants of the corresponding genes, around 1 kb upstream and downstream fragments of the targeted genes were amplified from *P. fici* and *A. fumigatus* genomic DNA (gDNA). The selected marker genes (*hph, neo, bar, AfpyrG*) were amplified from the vectors pUCH2-8, pAG1-H3-G418, dsPCNA-BAR, and pYH-WA-pyrG using the designed primers, respectively. The deletion

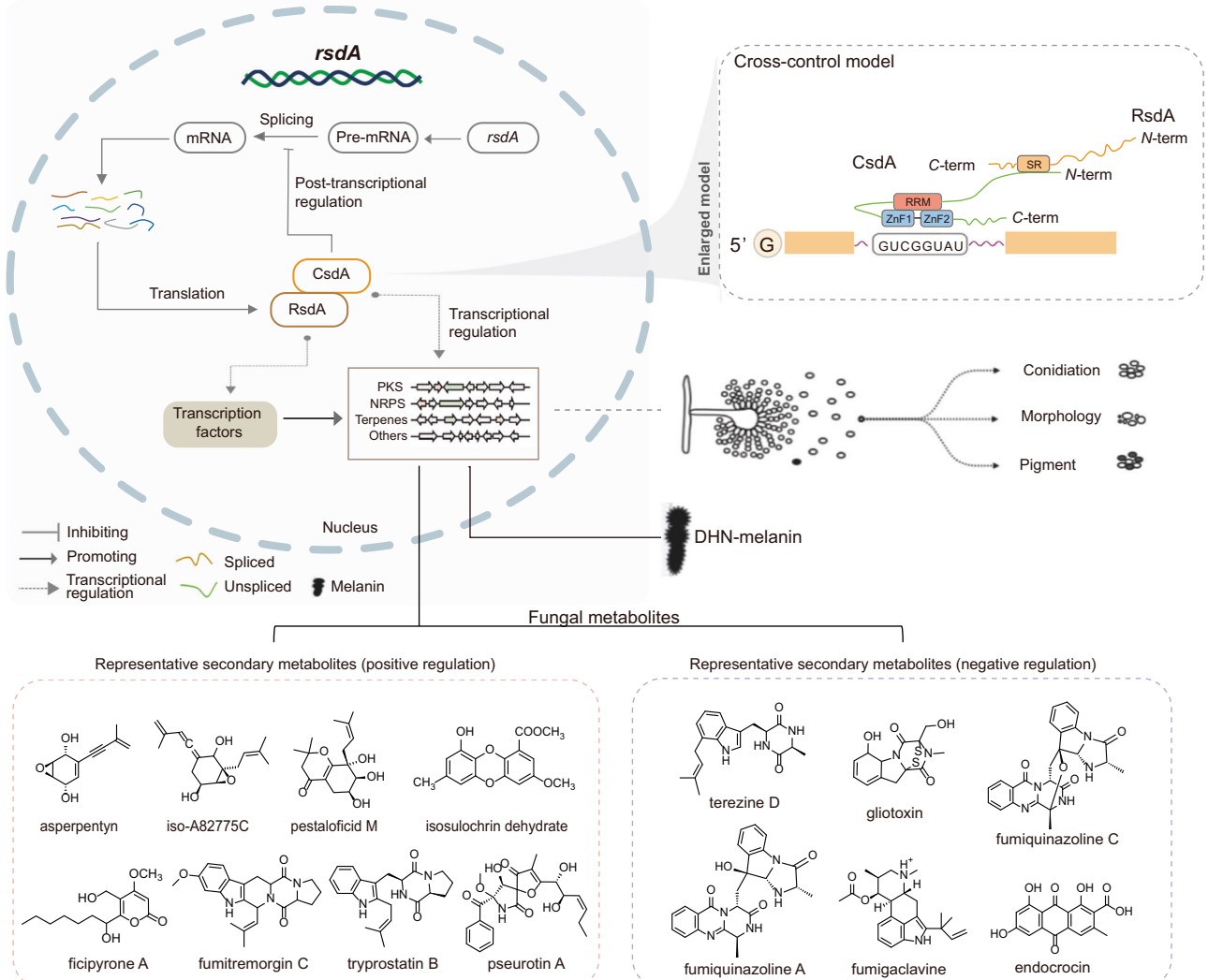

**Fig. 8 | The model shows that CsdA/RsdA interaction with each other to coordinate fungal secondary metabolism and development.** As an RNA-binding protein, CsdA acts as a splicing repressor of *rsdA* intron 2 at post-transcriptional modification, and CsdA is thought to contribute to the maintenance of RsdA expression in homeostasis. In the biosynthesis of secondary metabolites, the CsdA/ RsdA complex co-localizes in the nucleus and is considered as an upstream regulatory complex, which can regulate the synthesis of various SMs, including polyketides, non-ribosomal peptides, and terpenes, especially the synthesis of polyketide DHN-melanin.

cassettes of the targeted genes were constructed by homologous recombination in the double joint method[60]. For protein purification, the open reading frames (ORFs) of targeted genes were amplified from the cDNA of *P. fici* and inserted into pET28a (*His₆*-tag) or pGEX-4T (*GST*-tag) to produce pYZS42 (RsdA), pYSZL42 (RsdA$^{ΔC-terminal}$), pYSZL6 (CsdA), pYSZL43 (CsdA$^{ΔN-terminal}$), pYSZL44 (CsdA$^{ΔRRM}$) and pYSZL45 (CsdA$^{ΔZnF}$) through the quick-change method[60]. In order to determine the subcellular localizations of RsdA or CsdA, *rsdA* or *csdA* gene was amplified from the gDNA of *P. fici*, respectively. *sfGFP* and *mCherry* were amplified from corresponding vectors above. *rsdA*, *sfGFP*, and *neo* were integrated into the SpeI/PmlI-cleaved pXW55 vector via the yeast recombination method[33] to give the plasmid pYZS33. The same method was used to construct other fluorescent expression vectors pYSZL8 (*csdA-mCherry*). All plasmids were verified by PCR with corresponding primers.

Bimolecular fluorescence complementation (BiFC) plasmids were created using the Clone Express® MultiS One Step Cloning Kit (Vazyme Biotech Co. Ltd, China) according to the standard manufacturer's instruction. The *hph* or *neo* gene was cloned into the KpnI/HindIII-cleaved pKNT-NYFP or pCX62-CYFP plasmid to give the empty vector pYSZL30 (*hph-nyfp*) or pYSZL31 (*neo-cyfp*),

respectively. Each gene of *PfrsdA* and *PfcsdA* was fused with $P_{gpdA}$ and integrated into pYSZL30 or pYSZL31 to obtain pYSZL37 or pYSZL34, respectively. All plasmids were verified by sequencing (SANGON BIOTECH Co. Ltd, CHINA).

**Fungal genetic manipulations**

Deletion of the corresponding genes in *P. fici* was achieved according to the CRISPR/Cas9 technique[37]. In brief, PAM loci of 10 target genes were predicted by a CRISPR design tool (version 5.4, http://www.e-crisp.org/E-CRISP/). Each deletion cassette and its corresponding sgRNA were co-transformed into *P. fici* Cas9 (TYPZ36.2) strain. For subcellular localization of RsdA or CsdA, the DNA fragment of *rsdA-sfGFP* or *csdA-mCherry* was amplified from pYZS33 or pYSZL8, and transformed into *P. fici* Cas9 to produce strain TYZS41 or TYSZL24, respectively. For BiFC assays, each pair of plasmids (pYSZL37 and pYSZL31, pYSZL30 and pYSZL34, pYSZL37 and pYSZL34) were co-transformed to *P. fici* WT to produce strains TYSZL40, TYSZL39, and TYSZL34, respectively. In *A. fumigatus*, the deletion cassette of *AfrsdA* or *AfcsdA* was transformed into *A. fumigatus* CEA17.2 to generate TYYJ14 or TYSZL12, respectively[11]. Subsequently, to create a ΔAfrs-dAΔAfcsdA double mutant, the deletion cassette of *AfcsdA* with *hph*

gene was transformed to TYYJ14 to produce TYSZL14. All the above hygromycin B and G418 resistant colonies were verified by diagnostic PCR.

## Phylogenetic analysis of RsdA and CsdA involves species

The amino acid sequences of RsdA and CsdA from *P. fici* were used as the query for a BLAST analysis at the website of www.blast.ncbi. nlm.nih.gov/Blast.cgi, respectively. In total, 1474 species containing RsdA homologs and 1535 species containing CsdA homologs were retained. The phylogenetic tree was constructed based on the amino acid sequences of ZnF from 15 eukaryotes with MEGA 7.0 using the neighbor-joining method. The *lsu, ssu, rpb1, rpb2, its* and *ef1* genes from the identified 163 species were used to construct a phylogenetic tree[38]. The nucleotide sequences of *lsu, ssu, rpb1, rpb2, its* and *ef1* genes from 163 species were downloaded from the NCBI database (https://www.ncbi.nlm.nih.gov/), aligned with MEGA6.06 software, and manually adjusted. Sequence Matrix 1.7.8 was used to splice *lsu, ssu, rpb1, rpb2, its* and *ef1* genes of each strain in 163 species to construct the phylogenetic tree by RAxML 1.3.1 software[61]. The nucleotide substitution model was set as the default GTRGAMMA and clustering was performed by the maximum likelihood (ML) method, while the other parameters were default. The strains of Mucoromycota and Basidiomycota were regarded as the outgroup, and the reliability of internal branch was evaluated with 1000 bootstrap resampling. The phylogram was viewed and modified using the Interactive Tree of Life (ITOL, version v6, http://itol.embl.de/).

## Phenotypic analysis of different strains

*P. fici* and its mutant were cultured on a 25 °C PDA for 5 days for analysis of growth and 14 days for analysis of conidial production. The conidia were collected with 0.1% Tween-80 and counted by a hemocytometer. Each experiment was conducted in five biological replicates.

## Extraction and HPLC analysis of secondary metabolites

To extract SMs, *P. fici* and its mutants were cultivated on PDA at 25 °C for 7 days. *A. fumigatus* CEA17.1 and its mutants were cultivated on GMM medium at 37 °C for 3 days, and then $1 \times 10^7$ spores were inoculated into 20 ml liquid GMM medium at 25 °C for 5 days. SMs were extracted repeatedly with ethyl acetate and evaporated under reduced pressure. The extracts were dissolved in 1 ml methanol (MeOH) and then analyzed on a Waters HPLC system (Waters e2695, Waters 2998, Photodiode Array Detector) using an ODS column (C18, $250 \times 4.6$ mm, Waters XTERRA®, 5 μm) with a flow rate of 1 ml min$^{-1}$. For the SMs analysis of mutants and their control strains, MeOH (A) and water with 0.1% (v/v) formic acid (D) was used as the solvent. Extracts were eluted with linear gradient from 20 to 100% (v/v) A in 20 min, washed with 100% (v/v) solvent A for 5 min, and equilibrated with 20% solvent A for 5 min. UV absorptions at 254 nm were illustrated.

## Extraction and quantification of fungal melanin

The melanin of *P. fici* and its mutants was separately extracted by alkali extraction[62]. In brief, 1 cm$^2$ discs were cut with a blade from the center of a growing fungal colony, boiled in distilled water for 5 min, cooled, and washed three times. Subsequently, discs were autoclaved in 3 ml of 1 M KOH (20 min, 121 °C, 105 kPa). The mixtures were cooled and centrifuged at 4 °C and 13,000 × g for 10 min. The supernatants were mixed with 3 ml of 0.1 M borate solution (pH = 8.0). The concentration of melanin was measured spectrophotometrically (UNICO, China) at 540 nm. Standard curves were drawn by synthetic melanin (SIGMA, USA) with the concentrations ranging from 0 to 0.15 mg ml$^{-1}$. All data were repeated three times.

## Expression and purification of proteins

*E. coli* BL21 cells were transformed with the plasmid pYZS42 or pYSZL42 to express RsdA or RsdA$^{\Delta C\text{-terminal}}$ protein. Cells were cultivated at 37 °C in LB medium containing 50 μg ml$^{-1}$ kanamycin to an OD$_{600}$ of 0.6. Then, the expression of RsdA or RsdA$^{\Delta C\text{-terminal}}$ was induced with 0.5 mM isopropyl β-D-1-thiogalactopyranoside (IPTG) at 16 °C for 20 h. The cells were harvested and lysed by freeze-thaw, and cell debris was removed by centrifugation. Recombinant RsdA or RsdA$^{\Delta C\text{-terminal}}$ protein was purified with Ni-NTA resin (QIAGEN, CA)[21]. The expression and purification of CsdA, CsdA$^{\Delta N\text{-terminal}}$, CsdA$^{\Delta RRM}$, or CsdA$^{\Delta ZnF}$ protein were the same as those of RsdA, but the lysis buffer (50 mM Tris, 150 mM NaCl, 1 mM DTT, pH 7.3), GST-tag resin (Beyotime, China) and elution buffer (50 mM Tris, 150 mM NaCl, 10 mM GSH, pH 8.0) were different. The molecular weight and concentration of proteins were determined by 12% SDS-PAGE and Nano-Drop C2000 (Thermo Fisher Scientific), respectively.

To extract the total soluble proteins of *P. fici*, the mycelia grown at 25 °C for 5 days were harvested. The mycelia were ground into powder in liquid nitrogen and the power was loaded into centrifuge tubes. The powder was dissolved in an appropriate protein extraction buffer (137 mM NaCl, 50 mM HEPES, 10% Glycerol), Pepstatin A (1 g ml$^{-1}$), Leupeptin (1 g ml$^{-1}$), and PMSF (1 mM). The mixtures were incubated on ice for 10 min, then centrifuged at 4 °C and 13,000 × g for 15 min to obtain the total soluble proteins of *P. fici*. Total soluble proteins were detected by 12% SDS-PAGE.

## Pull-down assays

To screen for interacting proteins of RsdA, total soluble proteins of *P. fici* and recombinant RsdA protein were incubated in vitro. In brief, 1 mg of RsdA-His protein was loaded onto Ni-NTA resin, and 5 ml wash buffer (50 mM NaH$_2$PO$_4$, 300 mM NaCl, 30 mM imidazole, pH 8.0) was used to elute the unbound proteins, repeating the step three times. Ten mg of total soluble proteins of *P. fici* were then added to the above protein-resin mixture, and the unbound proteins were eluted again with 5 ml wash buffer. The bound protein mixtures were eluted with 3 ml elution buffer (50 mM NaH$_2$PO$_4$, 300 mM NaCl, 250 mM imidazole, pH 8.0) and detected by 12% SDS-PAGE. Entire lanes or four different electrophoretic bands were cut for liquid chromatography-mass spectrometry (LC-MS). Mascot software ($p < 0.05$) was used to identify the peptides. The two control samples (only RsdA or total soluble proteins of *P. fici*) were identified as described above.

To determine the interaction between RsdA and CsdA in vitro, recombinant RsdA or CsdA protein was incubated with Ni-NTA resin in binding buffer (50 mM Tris, 250 mM NaCl, 0.2% glycerol, 5 mM DTT, 0.6% Triton X-100, 1 mM PMSF, pH 7.5) at 4 °C for 4 h. The two mixtures were centrifuged at 4 °C and 800 × g for 1 min, and the supernatant of CsdA was added to the precipitate of RsdA and incubated overnight at 4 °C. Then, the CsdA/RsdA-resin mixtures were centrifuged at 4 °C and 800 × g for 1 min. The precipitate was then resuspended in wash buffer (50 mM Tris, 250 mM NaCl, 1% Triton X-100, 1 mM PMSF, pH 7.5). The centrifugation and resuspension were repeated three times. The mixtures were detected by western blotting. In vitro pull-down of RsdA and CsdA$^{\Delta N\text{-terminal}}$, CsdA$^{\Delta RRM}$, or CsdA$^{\Delta ZnF}$ and RsdA$^{\Delta C\text{-terminal}}$ and CsdA followed the same procedure as above.

## Western blotting

The mixtures of CsdA/RsdA-resin were incubated at 100 °C for 10 min. CsdA and RsdA proteins were separated by 7.5% SDS-PAGE. The isolated proteins were then transferred to polyvinylidene fluoride (PVDF) membrane (PALL, USA). His- and GST-tagged proteins were detected with monoclonal anti-His (Proteintech, Chicago, USA, 1:10,000 dilution) and anti-GST (Proteintech, Chicago, USA, 1:10,000 dilution) antibodies, respectively. HRP-Goat Anti-Mouse IgG (Proteintech, Chicago, USA, 1:5000 dilution) was used to hybridize with His and GST

antibodies, respectively. The bands were visualized by ECL (Thermo, MA, USA).

## Transcriptional analysis by RNA-seq and qRT-PCR

*P. fici* and its mutants were cultivated on PDA at 25 °C for 5 days. Three replicates were performed for each strain. Total RNAs from the mycelia of *P. fici*, Δ*rsdA*, and Δ*csdA* mutants were isolated using a TranZol™ kit (TransGen Biotech, China). Then the quality of RNA was checked with an Agilent 2100 bioanalyzer. For RNA-seq analysis, RNA examples were sequenced on an Illumina NovaSeq 6000 (Illumina, USA) at Novogene Biotech Co. Ltd. Data quality was controlled by fastp (version 0.19.7)[63], and the sequencing error rate for a single base location was less than 1%. The clean reads were mapped to the reference sequence by HISAT2[64], and the Integrative Genomics Viewer (IGV, version 2.12.3) was used to visualize the mapped reads number. Gene expression levels were calculated using normalized FPKM (fragments per kilobase of transcript per million mapped reads). The differentially expressed genes were calculated by DESeq2[65] and identified with *p* value and log$_2$foldchange/ratio between *P. fici* and mutants.

The expression levels of specific genes (*pfma* and *iac* cluster, *rsdA*, and *csdA* genes) in *P. fici* and mutants were verified by quantitative real-time PCR (qRT-PCR). Total RNAs of *P. fici*, Δ*rsdA*, and Δ*csdA* were reverse transcribed into cDNA by *Evo M-MLV* Plus cDNA Synthesis kit (Accurate Biotechnology(Hunan) Co. Ltd, China) according to the manufacture's protocol. qRT-PCR was conducted using a CFX96 Real-Time System (BIO-RAD). A KAPA SYBR FAST qPCR Kit (Kapa Biosystems, USA) was used for the reactions (2 × KAPA SYBR FAST qPCR Master Mix, 0.2 µM forward/reverse primer, cDNA template ~2 µg, RNA-free water to 20 µl). The reactions were carried out at 95 °C for 3 min, followed by 40 cycles of (95 °C for 3 s, 60 °C for 20 s, 72 °C for 20 s). Each cDNA sample was performed in triplicate and the average threshold cycle was calculated. Relative expression levels were calculated using the $2^{-\Delta\Delta Ct}$ method[33]. qRT-PCR primers are listed in Supplementary Data 8.

## Metabolome determination and analysis

To investigate the comprehensive changes of metabolites caused by RsdA and CsdA, the metabolome of *P. fici* and its mutants were determined by LC-HRMS (Agilent HPLC 1200 series system)[66]. *P. fici*, Δ*rsdA*, and Δ*csdA* strains were cultivated on 20 ml PDA at 25 °C for 7 days. The metabolites were extracted with 40 ml of an ethyl acetate/methanol mixture (9:1, v/v) and sonicated for 1 h. The organic layer was evaporated to dryness, the crude extract was re-dissolved in 1 ml MeOH, and then analyzed by LC-HRMS using an ODS column (C18, 250 × 4.6 mm, Waters XTERRA®, 5 µm) with a flow rate of 1 ml min$^{-1}$. Linear gradient conditions of the mobile phase were as follows: 10%–30% MeOH (0–10 min), 30%–70% MeOH (10–40 min), 70%–90% MeOH (40–50 min), 100% MeOH (50.1–60 min), 10% MeOH (60.1–65 min). The molecular weights were scanned between 80 - 2000 *m/z*. All the LC-HRMS data were extracted using Agilent Masshunter software (version B.06.00) and further converted to the format containing retention time, *m/z*, and ion peak density via XCMS website (https://xcmsonline.scripps.edu). PCA and enrichment analysis were performed using the OmicStudio tools (version 3.6) at https://www.omicstudio.cn/tool. Ion product summaries and volcano maps were drawn with GraphPad Prism (version 8.0) software. Differential metabolites were screened with |Log$_2$foldchange| > 1 and −Log$_{10}$ (*p* value) > 1.3. The metabolome of *A. fumigatus* CEA17.1, Δ*AfrsdA*, Δ*AfcsdA*, and Δ*AfrsdA*Δ*AfcsdA* strains were determined by the same method as above.

## Fluorescence detection and BiFC assays

TYZS41 (*rsdA-sfGFP*) or TYSZL24 (*csdA-mCherry*) in the background of *P. fici* Cas9 was activated and cultivated on PDA for 4 days at 25 °C for subcellular localization of RsdA or CsdA. The mycelia were then picked into PDB for further cultivation for 5 days and harvested by centrifugation. The collected mycelia were fixed with 10% formalin for 30 min and washed with distilled water. DAPI solution (final concentration: 10 µg ml$^{-1}$, Biosharp, China) was incubated with mycelia for 15 min and washed with distilled water. Fluorescent signals were observed with a fluorescence microscope. Images were acquired using a Zeiss Axioplan 2 imaging system with the AxioCam MRm camera (Carl Zeiss Microscopy). Digital images were taken and processed with the IMAGEJ2 software (National Institutes of Health)[49].

To confirm the interaction between RsdA and CsdA in vivo, the NYFP- or CYFP- tagged BiFC strains were detected by fluorescence microscopy. TYSZL40, TYSZL39, and TYSZL34 in the background of *P. fici* WT were cultivated and stained according to the fluorescence detection method of *P. fici* described above. All images were taken and processed with IMAGEJ2 software (version 2.15.0).

## Structure prediction and docking of proteins

To confirm the function of CsdA, the protein structure of CsdA containing 730 amino acids was predicted by AlphaFold2[44]. PfRRM or PfZnF domains were used as queries to perform multi-sequence alignment with the reported domains in the PDB database (https://www.rcsb.org/) by MEGA7 software. The dimer structure of RsdA and CsdA was simulated by AlphaFold2. The pre-mRNA structure of *PfrsdA* was predicted by 3DRNA (version 2.0)[67]. The CsdA protein docked with ~500 bp of pre-mRNA around intron 2 of the *rsdA* by HADDOCK 2.4[68]. Topology analysis and conserved domain annotation were performed with the PyMOL2.5 software package. All parameters were the default settings.

## Measurement of splicing ratio

Total RNAs extraction and cDNA synthesis of *P. fici*, Δ*rsdA*, Δ*csdA*, and CsdA$^{R368A-F373A}$ were performed as described above. The splicing ratio was measured by qRT-PCR method[49]. In brief, the splicing ratio was calculated by normalizing the spliced RNA abundance to the unspliced RNA abundance of each intron using the equation $2^{\Delta (Ct-unspliced - Ct-spliced)}$. The splicing ratio of *P. fici* was normalized to 1. To amplify the spliced RNA, the forward primer was designed to cross exon-exon junctions, and the reverse primer was designed to be in the exon adjacent to the intron (Fig. 4b). To amplify the unspliced RNA, forward primer was designed in the intron, and reverse primers were identical to the spliced RNA (Fig. 4b). The corresponding primers are listed in Supplementary Data 8.

## Electrophoretic mobility shift assay (EMSA) and isothermal titration calorimetry (ITC) assays

For determination of the binding ability of CsdA with the potential element GUCGGUAU, the T7 promoter was fused to the 5′ end of 5′-CCAGGGAGGTGAAGAAGTCGGTATGTTTCACCAGATGTGAT-3′, and then RNA containing GUCGGUAU was prepared with a T7 High Efficiency Transcription Kit (TransGen Biotech, China). The prepared RNA was purified by ssDNA/RNA Clean & Concentrator™ Kit (Zymo Research, USA) and biotin-labeled by EMSA Probe Biotin Labeling Kit (Beyotime, China). In vitro binding of recombinant protein GST-CsdA to RNA was performed via the EMSA/Gel-Shift Kit (Beyotime, China) following the manufacturer's instruction. Briefly, purified GST-CsdA was mixed with biotin-labeled RNA, and then inoculated for 20 min at 25 °C. The reaction mixtures were separated on a 6% polyacrylamide gel for 90 min, and then transferred to a positively charged nylon membrane (Beyotime, China). The signals were detected by western blotting. Fifty nt RNAs around the 5′ splicing sites of introns 1 and 3 served as negative controls.

In order to further confirm the pre-mRNA target of CsdA, 4 × GUCGGUAU (5′-GTCGGTATCGTCGGTATGGTCGGTATTGTCGG-TAT-3′) was designed and synthesized according to the same method. Isothermal Titration Calorimetry (ITC) assays were

performed by using AFFINITY ITC LV (Waters, USA). For detecting the binding affinity, 200 μM 1 × GUCGGUAU or 4×GUCGGUAU was titrated into 50 μM CsdA at 25 °C. Data were analyzed using Launch Nano Analyze software (version 3.11.0). The experiment was repeated twice independently.

### Statistics and reproducibility

Statistical parameters are reported either in individual figure or corresponding figure legends. Quantification data are generally presented as bar/line plots, with the error bar representing mean ± SD. All statistical analyses were done with GraphPad software (version 8.0). Asterisks were used to indicate statistical significance, * stands for $p < 0.05$; **$p < 0.01$, ***$p < 0.001$, and ****$p < 0.0001$. Representative images from two independent experiments showed similar results in Fig. 1b and Supplementary Fig. 1a, b. The phenotype, micromorphology and fluorescence of the strains were examined in at least three independent experiments in Figs. 1e, 2c, 3a, 4b and Supplementary Fig. 1e. Representative images from two independent in vitro pull down or in vivo BIFC experiments showed similar results in Fig. 3b, c, e. EMSA assays were carried out two times and representative images are shown in Fig. 5f.

### Reporting summary

Further information on research design is available in the Nature Portfolio Reporting Summary linked to this article.

## Data availability

The data supporting the findings of this study are available from the corresponding authors upon request. RNA-seq data generated in this study have been deposited in the Gene Expression Omnibus (GEO) database under accession code GSE241019. Metabolomics data generated in this study have been deposited in the MassIVE database under accession code MSV000091220. Source data are provided with this paper.

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

## Acknowledgements

We thank Professor Zhonghua Ma from Zhejiang University for providing the bimolecular fluorescent complementary vectors. We thank Dr. Wenzhao Wang from Institute of Microbiology, Chinese Academy of Sciences for metabolome data collection. This work was supported by the National Key Research and Development Program of China [grant no. 2022YFC2303000]; the National Natural Science Foundation of China [grant no. 32170066]; the Key Research Program of Frontier Sciences, Chinese Academy of Sciences [grant no. ZDBS-LY-SM016]; the Biological Resources Program, Chinese Academy of Sciences [grant no. KFJ-BRP-009-005]; and the Strategic Priority Research Program of the Chinese Academy of Sciences [grant no. XDA28030402].

## Author contributions

W.-B.Y. designed the research and supervised the study. Z.S. performed fungal phenotypic analysis, fermentation, HPLC analysis, transcriptome and metabolome analysis, phylogenetic analysis, subcellular localization, pull-down assay, BiFC assay, protein structure prediction, conserved domain analysis and qRT-PCR experiment. S.Z. carried out in vitro protein binding assay and mass spectrometry identification of RsdA and *P. fici* total proteins. H.Z. performed amino acid mutations in fungi and analyzed the mutant's phenotype and secondary metabolites. X.L. supervised the in vitro assay. N.P.K. and B.R.O. revised the manuscript. Z.S. and W.-B.Y. wrote the paper.

## Competing interests

The authors declare no competing interests.
