## [Peer Review File · Nature Communications]

Fungal secondary metabolism is governed by an RNA-binding protein CsdA/RsdA complexREVIEWER COMMENTS

Reviewer #1 (Remarks to the Author):

As far as I am aware this is the first published report of an RNA binding complex (RsdA/CsdA) that contributes to 'global' regulation of genes for secondary metabolism (SM). This is a seminal discovery because, although we know that associated genes are plentiful in fungi, thanks to genomics, nailing down when the very diverse SM genes are expressed has been a long sought-after goal. The 'holy grail' in this endeavor has been to find genes involved in global regulation of all or most. Correspondingly, lack of knowledge of where in the life cycle of the fungus these genes are expressed has made determination of function and identification of corresponding metabolites very tedious not to mention challenging. One size does not fit all for these diverse gene clusters and hence the hunt for 'global' regulators (regulators that control expression of many BCGs at once). Although the RsdA function was published already, the addition here is CsdA which, as noted above, is complexed with RsdA. This is an original and important discovery related to global regulation and because these genes are conserved across the fungi, likely to be a general mechanism.

The work is very thorough. Employs comprehensive methodology to reach solid conclusions converging from several angles (genetics, gene knockouts, transcriptome comparisons, phenotypic comparisons between mutants and wild type biochemical comparisons between mutants and wild type, localization studies and even binding sequence). Nicely summarizes each section of Results.

My main criticism is the manuscript could use careful editing of the English throughout to tighten up descriptions and eliminate grammatical errors.

'Global' is not global but rather controls other SMs as well. I can't think of a better term though.

Misc:

27. Microbial should be microbial

28-34. please acknowledge that SMs have many functions both for the fungus itself and in interactions with other organisms as well as the more serendipitous pharmaceutical/medicinal roles.

38-39. I don't understand this sentence 'in nature...'

40-42. Needs greater clarity since this is the heart of the rationale.

47. be careful of generalizations – mostly restricted to a few fungi (eg Aspergilli)

48-60. Old literature please add recent updates.

62. delete 'together' -assemble is sufficient

158. analysis among ... change to 'analysis of'....

298. An example of poor sentence structure – needs rewriting especially since this is a major finding.

317. Is 'specifically' the correct word here? Also confusing – percentages or percentage, were or was?

Figure legends. In some cases, could be tightened up --descriptions of methodology taken out since they are in M&Ms

Reviewer #2 (Remarks to the Author):

The manuscript by Song et al. describes the regulation of secondary metabolism in the endophyte *Pestalotiopsis fici*. The authors discover that the RNA-binding protein CsdA is involved. The RBP interacts with the global regulator RsdA and it has an influence on the splicing of the RsdA encoding mRNA. The authors conclude that the complex regulates secondary metabolism.

The important role of RNA-binding proteins in the regulation of fungal biology is an emerging theme and currently only few examples are described. Therefore, this study addresses a timely research question. However, the experimental evidence for their claims e.g. that the complex is a global regulator for secondary metabolism is not solid enough to justify publication in Nature Communications.

Major point:

1. Based on their RNAseq data 720 mRNAs are co-regulated suggesting that both proteins function as complex. The genetic analysis of the deletion mutants revealed separate functions. In both cases the growth rate is reduced. However, the conidial development is clearly different. Furthermore only 39.8% of differentially regulated metabolites overlap. It is conceivable that one regulator inhibits the function of the other. But this scenario has to be demonstrated by additional experiments.
2. The authors show convincingly, that the two proteins bind to each other. However, the function of this interaction is unclear. To address this point it would be, for example, very informative to express mutant proteins that are unable to interact in the fungus and test for the changes in development and secondary metabolism. The authors show the potential interaction interphase of CsdA with RsdA at the amino acid level using AlphaFold2. It is important to narrow down the interaction interface also experimentally. Thereby, the authors could identify mutants that no longer interact.
3. The same holds true for the RNA interaction. The authors demonstrate binding to the sequence GUCGGUAAU. Here, a negative controls of similar RNA sequences is missing to demonstrate sequence specificity. To demonstrate that this interaction is functional meaningful, the CsdA binding site should be mutated in the background of the RsdA gene. According to the authors this should result in changes in alternative splicing and the resulting influence on development and secondary metabolism could be tested. Alternatively, constitutive expression of the alternatively splice version of *rsdA* might rescue the phenotype.
4. The role of CsdA in splicing needs to be tested transcriptome-wide. The RNA seq data of CsdA deletion strain are available and should be evaluated, for example but studying exon/exon spanning reads, that report the rate of splicing. It is very rare, that a RNA-binding protein has only a single target mRNA. In fact, the RNAseq experiments demonstrate that 1262 genes are differentially regulated in CsdA deletion mutants. 720 genes are co-regulated indicating that a substantial amount of mRNAs alter their expression independent of RsdA. Thus, one would expect more targets for such a global regulator.

Minor points

1. Introduce RsdA in more detail. What type of global regulator is RsdA ? A DNA-binding transcription factor ?
2. In line 356, the lane number are not given correctly.
3. Line 370-373, what does n=4 indicates? If it is number of experiments, then the three replicates

statement in the line 381 indicates the technical replicates?

4. In figure 4B, the arrow marks to indicate primers is not clear. It would be clearer if the authors could specify the reasons for different arrow colours (blue vs red) in the figure legend.

5. In the model, Fig 6C, the domain architecture of RRM and ZnF rather seems to be disconnected into two parts.

6. In line 88, the argument to identify protein interaction partners of RsdA seems poor.

7. While the text in line 148 suggests that in total of 2218 (39.8%) of metabolite ions were co-regulated by CsdA and RsdA, the correct number according to the following text and Fig. 2F is 1333.

8. The authors have used purified, recombinant RsdA to pull down interaction partners from the total cell lysate. It is unclear to me why they didn't use the in vivo RsdA for the pull-down experiment.

9. In the methods section for "Transcriptional analysis by RNA-seq and qRT-PCR", the initial quality control steps used for RNA-seq read processing is missing. This section also misses the citation for methods such as HISAT and DESeq 2 analysis.

10. The PCA analysis in FigED.1I showed high principal component variation between *csdAD* and *rsdAD* cells for the metabolite changes. If these proteins functions together in secondary metabolism, one might expect less variation as it has been observed for *A. fumigatus* in Fig 6B?

Reviewer #3 (Remarks to the Author):

The manuscript by Song et al. reports identification of the RNA-binding protein CsdA and posits that CsdA acts as a global regulator of fungal secondary metabolite production, in conjunction with another protein, the recently identified RsdA. In this current manuscript, the authors use pull-down in combination with other techniques to identify proteins that physically interact with RsdA. Among 10 candidates, one – CsdA – influences fungal development and small molecule production in a similar way as RsdA, and thus the authors then focus on CsdA. Very interestingly, it is then shown that RsdA and CsdA interact in the nucleus, and that CsdA may interact with the mRNA of RsdA (but see comment below).

Further it is shown that CsdA is widely conserved in filamentous fungi (usually with RsdA), and that deletion of either gene in diverse fungi strongly disrupts development, causes similar transcriptomic changes, and results in starkly altered metabolic profiles.

However, the principal claim of this paper – that CsdA (and RsdA) are regulators of global secondary metabolite production, is not supported by any of the presented data. CsdA (and RsdA) mutants have severe morphological defects and grow much slower, for example, it is demonstrated that both RsdA and CsdA play key roles in growth and conidial formation as well as pigmentation of *P. fici*. It is well established that almost any change in development affects secondary metabolite production. The strong growth, conidial, and pigmentation phenotypes observed indicate that development is seriously disrupted, and of course the spectrum of associated secondary metabolites will also be very different compared to WT. Similarly, many genes are changed in CsdA (and RsdA) mutants, and, unsurprisingly,

among them are secondary metabolite gene clusters. Importantly, there's no evidence that either of the two genes specifically affects metabolism.

While I don't think the claim that CsdA is "a global regulator of secondary metabolism" makes sense, the work is very interesting – I wonder about the role of CsdA (and RsdA) in fungal development, and a broader exploration of the regulatory pathways controlling expression of CsdA (and RsdA) would likely reveal key insights in the control of fungal development. I recommend that the authors consider reframing their work, focusing on the role of CsdA (and RsdA) in overall development, which necessarily will also reveal how secondary metabolite production ties in.

Minor points:

- homology of CsdA (in fungi and across eukaryotes) must be discussed more clearly, including the finding that the fungal RRM includes the ribonucleoprotein domains 1 and 2 (in line 231 it says, incorrectly, "ribonucleoproteins 1 and 2 (RNP1, RNP2)").
- the hypothesis that CsdA binds to the pre-mRNA of *rsdA* is exciting, but there seems to be a lack of controls in the experiments described in the section starting in line 245. I understand that docking suggested that CsdA binds to the GUCGGUUAU motif, but for the following electrophoretic mobility shift assay results to be meaningful, I think it would be necessary to include other control sequences, ideally from the same pre-mRNA. Moreover, the binding affinities don't seem very high (micromolar) – here it would help to give some perspective, e.g. by comparing these values with binding data for other protein-mRNA complexes.
- the introduction is overly long and not focused. For example, lines 38-40 contain trivial statements about basic ecology that seem unnecessary.

Reviewer #4 (Remarks to the Author):

"Fungal secondary metabolism is governed by an RNA-binding protein CsdA/RsdA complex" by Song et al.

Review Summary

Fungal secondary metabolites (SMs) have been rich sources of drug discovery, as exemplified by the historic drugs penicillin and lovastatin. However, a large portion of fungal SMs are still hidden, due to highly context-specific regulation of biosynthetic gene clusters (BGCs). In this manuscript, Song et al. tries to tackle this limitation by investigating the molecular action mechanisms of global regulator of SM, RsdA, which accounts for the regulation of more than 50% of BGCs throughout the filamentous fungi species. The authors identified CsdA as a novel interactor of RsdA protein, and as an RNA-binding protein regulating alternative splicing of *rsdA* pre-mRNAs as well. Comprehensive transcriptomics and metabolomics analysis revealed that RsdA and CsdA are functionally intertwined to co-regulate a large portion of secondary metabolites and BGCs (39.8 % of metabolic ions and about 24 key BGC backbone

genes out of 76 genes).

Overall, this manuscript provides novel mechanistic insights for the global regulation of fungal SMs which may open up new possibilities of discovering many hidden SMs that might be useful for new drug/chemical screenings. Therefore, I believe the study is of interest to broad audiences from microbiology, RNA biology, drug discovery and agriculture. However, at the same time, the conclusions are often based only on weak experimental evidence or from simple speculations, thus not yet strong enough, especially for being published in Nature communications. I would suggest 3 major points and a few minor suggestions for the authors to revisit to make the manuscript more conclusive and useful to wide audiences.

Major points

1. The functional significance of CsdA-RsdA protein complex:

The authors identified CsdA as a new interactor of RsdA and proposed it as a regulator of RsdA in two folds: RsdA protein function and *rsdA* pre-mRNA splicing. Unlike the pre-mRNA part, the functional significance of CsdA-RsdA protein complex in coordinating secondary metabolism is not clearly demonstrated, even though the authors show direct protein-protein interaction and nuclear co-localization between CsdA and RsdA. Likewise, the functional significance of CsdA-RsdA protein interaction in either the CsdA's role as an RBP or the RsdA's role as a global TF regulator are not thoroughly investigated. Therefore, it is still unclear to me whether the CsdA-RsdA protein complex has any functional roles in secondary metabolism or BGC regulation. I would suggest for the authors to perform some more experiments to address these concerns. Below are some questions and suggestions related to the specific point, but the authors can choose any other experiments to ultimately demonstrate the functional significance of CsdA/RsdA protein complex in secondary metabolism and BGC regulation.

- What are the changes of SMs (or BGC changes) in delta-*csdA*/delta-*rsdA* double mutant as compared with single delta-*rsdA* or single delta-*csdA* mutant?
- Can the CsdA point mutant which is deficient in RsdA interaction be generated by considering the structural modeling in Fig3d? For example, by mutating H26 and/or Y29 to A and/or F? If so, then can this mutant still be able to rescue delta-*csdA* phenotypes?
- Can the delta-*csdA* phenotypes be rescued by the complementation of *rsdA* mature mRNA? If *rsdA* mature mRNA can fully rescue delta-*csdA* phenotypes, then one may conclude that CsdA's role in secondary metabolism and fungi growth is mainly via alternative splicing of *rsdA* pre-mRNA. On the other hand, if this is not the case, CsdA must have additional roles in modulating RsdA protein function or RsdA-independent roles in the secondary metabolism.
- For CsdA to function as a splicing factor, it should first be in the nucleus but CsdA has no (putative) NLS. Is the nuclear localization of CsdA dependent on its interaction to RsdA? This can simply be tested by analyzing CsdA localization in delta-*rsdA* background.

2. What are the conditions (e.g., environmental stresses or upstream signalings) in which CsdA-RsdA expressions/activities are regulated (increased or decreased) for fungal growth and SMs generation? For example, can LaeA activate CsdA-RsdA?

3. What will be the potential applications of the findings (e.g., specific inhibitors or activators acting on CsdA-RsdA interface for inducing novel SMs)? Can the authors at least perform some proof-of-concept experiments for possible applications?

Other minor points

1. It is not yet clearly demonstrated that the RsdA-CsdA protein complex itself plays important roles in fungal secondary metabolism. The manuscript title and other texts in the abstract and discussion should be toned down unless this would be confirmed by further experimentations. The model in figure 6c should also be corrected.

2. Throughout Fig1 and 2, delta-csdA and delta-rsdA show similar but different phenotypes in terms of growth and secondary metabolites. Can these phenotypes be explained by DEGs from RNA-seq? It would be nicer to show whether the co-regulated BGC backbone genes in Fig2g are more related to the coregulated metabolites in Fig 2e-f. More generally, do the BGC backbone genes regulated by CsdA and/or RsdA show correlation with the down or upregulated metabolic ions by CsdA and/or RsdA shown in Fig 2e-f? Do the BGC genes (or other DEGs) that are oppositely regulated by CsdA and RsdA regulate conidial formation and may explain the differential phenotypes?

3. How the alternative splicing of rsdA affects rsdA mature mRNA level? Would intron2 inclusion increase rsdA mRNA stability? What would be the consequences of intron2 inclusion or exclusion to the final RsdA protein sequence, if any?

4. Would it be possible to analyze global splicing in a more unbiased and quantitative way from RNA-seq data instead of simply cherry-picking some genes of interests (as in Fig4 and 5)? The authors may find more target genes that are alternatively spliced by CsdA and regulate secondary metabolism.

5. Where are the binding sites of CsdA (GUCGGUAAU) located in the rsdA gene? Are there single or many GUCGGUAAU motifs and are they enriched near exon2-intron2 junction, intron2-exon3 junction or other sites? What would be the possible molecular mechanisms of CsdA in regulating splicing in terms of its possible interactions with the components of spliceosome?

Reviewer 1:

Comments:

As far as I am aware this is the first published report of an RNA binding complex (RsdA/CsdA) that contributes to 'global' regulation of genes for secondary metabolism (SM). This is a seminal discovery because, although we know that associated genes are plentiful in fungi, thanks to genomics, nailing down when the very diverse SM genes are expressed has been a long sought-after goal. The 'holy grail' in this endeavor has been to find genes involved in global regulation of all or most. Correspondingly, lack of knowledge of where in the life cycle of the fungus these genes are expressed has made determination of function and identification of corresponding metabolites very tedious not to mention challenging. One size does not fit all for these diverse gene clusters and hence the hunt for 'global' regulators (regulators that control expression of many BCGs at once). Although the RsdA function was published already, the addition here is CsdA which, as noted above, is complexed with RsdA. This is an original and important discovery related to global regulation and because these genes are conserved across the fungi, likely to be a general mechanism. The work is very thorough. Employs comprehensive methodology to reach solid conclusions converging from several angles (genetics, gene knockouts, transcriptome comparisons, phenotypic comparisons between mutants and wild type biochemical comparisons between mutants and wild type, localization studies and even binding sequence). Nicely summarizes each section of Results.

Thank you very much for your positive comments. According to your suggestions, we have rearranged some contents in the revised version. Below, we will make detailed response to each comment.

My main criticism is the manuscript could use careful editing of the English throughout to tighten up descriptions and eliminate grammatical errors.

'Global' is not global but rather controls other SMs as well. I can't think of a better term though.

Thank you very much for your suggestion. The language of the manuscript has been carefully edited and polished by the native speakers of our co-authors Professors Nancy Keller and Berl Oakley.

Misc:

27. Microbial should be microbial.

We removed "The" in "The Microbial", so "Microbial" is correct.

28-34. please acknowledge that SMs have many functions both for the fungus itself and in interactions with other organisms as well as the more serendipitous pharmaceutical/medicinal roles.

In lines 28-34, we added the sentence as "Many of these compounds serve crucial functions for the microorganism itself and play significant roles in interactions with other organisms. Notable examples include the UV-resistant DHN-melanin (Zhang, P. et al., Mol Microbiol. (2017) 105(3):469-483) and the anti-fungal phenazine synthesized during bacterial-fungal interactions (Zheng, H. et al., Curr Biol. (2015) 25(1):29-37)".

38-39. I don't understand this sentence 'in nature...'

We removed the sentence from the manuscript to make the introduction more readable.

40-42. Needs greater clarity since this is the heart of the rationale.

To clarify this, we rephrased the sentence as “Gene regulation strategies have been effectively devised and implemented to activate dormant or low-expressed BGCs, enabling the exploration of novel fungal natural products (Lyu, H.N. et al., Nat Prod Rep. (2020) 37(1):6-16). Nonetheless, the intricate mechanisms underlying the regulation of SM genes in fungi remain only partially elucidated”.

47. be careful of generalizations – mostly restricted to a few fungi (eg Aspergilli)

We revised the sentence as “Among these known BGCs, 12 BGCs (42.3%) contain 16 PSTFs in *A. nidulans*, and 10 BGCs (55.6%) contain 12 PSTFs in *A. fumigatus* (Wang, W. et al., Int J Mol Sci. (2021) 22(16):8709)”.

48-60. Old literature please add recent updates.

The literatures 20 (Caceres, I. et al., Toxins (Basel). (2020) 12(3):150), 21 (Yin, W.B. et al., Mol Microbiol. (2012) 83(5):1024-34), 22 (Ries, L.N.A. et al., PLoS Pathog. (2020) 16(7): e1008645), 28 (Schönig, B. et al., Eukaryot Cell. (2008) 7(10):1831-46), 29 (Tannous, J. et al., Mol Plant Pathol. (2018) 19(12):2635-2650), 30 (Ke, R. et al., J Theor Biol. (2013) 326:11-20) in lines 48-60 as well as the full text have been updated.

62. delete ‘together’ -assemble is sufficient.

Done.

158. analysis among ... change to ‘analysis of’....

Done.

298. An example of poor sentence structure – needs rewriting especially since this is a major finding.

Thank you very much for your suggestion. We revised the sentence as “The unexpected revelation of this RBP's involvement in the comprehensive regulation of fungal secondary metabolism represents a significant discovery”.

317. Is ‘specifically’ the correct word here? Also confusing – percentages or percentage, were or was?

We revised the sentence as “In *P. fici*, it was observed that approximately 32% of the BGC backbone genes were found to be co-regulated by both CsdA and RsdA (Fig. 2g)”. Corrections addressing similar issues have been made throughout the entire text.

Figure legends. In some cases, could be tightened up--descriptions of methodology taken out since they are in M&Ms.

We removed the extra description in the full-text figure legends to refine it.

Reviewer 2:

The manuscript by Song et al. describes the regulation of secondary metabolism in the endophyte *Pestalotiopsis fici*. The authors discover that the RNA-binding protein CsdA is involved. The RBP interacts with the global regulator RsdA and it has an influence on the splicing of the RsdA encoding

mRNA. The authors conclude that the complex regulates secondary metabolism.

The important role of RNA-binding proteins in the regulation of fungal biology is an emerging theme and currently only few examples are described. Therefore, this study addresses a timely research question. However, the experimental evidence for their claims e.g., that the complex is a global regulator for secondary metabolism is not solid enough to justify publication in Nature Communications.

Thank you very much for your positive comments. According to your suggestions, we have rearranged some contents in the revised version. Below, we will make detailed response to each comment.

Major point:

1. Based on their RNAseq data 720 mRNAs are co-regulated suggesting that both proteins function as complex. The genetic analysis of the deletion mutants revealed separate functions. In both cases the growth rate is reduced. However, the conidial development is clearly different. Furthermore only 39.8% of differentially regulated metabolites overlap. It is conceivable that one regulator inhibits the function of the other. But this scenario has to be demonstrated by additional experiments.

Indeed, regulatory effects of CsdA and RsdA on conidial development and secondary metabolites (SMs) in *P. fici* were found to be intricate. To address this complexity, we performed a deletion of *rsdA* in the Δ *csdA* mutant background and observed that the conidial number of the Δ *csdA* mutant could be restored. The data would be consistent with CsdA and RsdA having independent and opposite effects on conidiation. Furthermore, we focused on the representative SM, pestheic acid, which exhibited contrasting regulation patterns by CsdA and RsdA. Notably, we discovered that the production of

pestheic acid in the $\Delta\text{csdA}\Delta\text{rsdA}$ mutant resembled that of the control strain, suggesting that the effects of the two deletions are independent and additive. Additionally, these results have been incorporated into the second part of the manuscript's results section (see please Supplementary Fig. 3). Furthermore, we observed that this phenomenon is widespread among fungi, as exemplified by the distinct regulation of asexual development in the VelB-VosA complex of *A. nidulans* (Park, H.S. et al., PLoS One. (2012) 7(9): e45935).

Supplementary Figure 3. Analysis of conidial number and secondary metabolites in the $\Delta\text{csdA}\Delta\text{rsdA}$ mutant of *P. fici*. **a**, Conidial production in *P. fici* and its mutant strains. **b**, HPLC analysis of SMs in *P. fici* and its mutant strains. UV absorptions at 254 nm are illustrated. **c**, Production of pestheic acid in *P. fici* and its mutant strains. All error bars are expressed as \pm SD. Statistical analysis was performed by using Two-way ANOVA (Significant at **** $p < 0.0001$).

2. The authors show convincingly, that the two proteins bind to each other. However, the function of this interaction is unclear. To address this point it would be, for example, very informative to express mutant proteins that are unable to interact in the fungus and test for the changes in development and secondary metabolism. The authors show the potential interaction interphase of CsdA with RsdA at the amino acid level using Alphafold2. It is important to narrow down the interaction interface also experimentally. Thereby, the authors could identify mutants that no longer interact.

We have done the experiments as you suggest. In order to investigate the protein interactions between RsdA and CsdA, we employed AlphaFold2 prediction and conducted *in vitro* pull-down assays. The analysis suggested that the C-terminal of RsdA and the N-terminal of CsdA play crucial roles in these interactions. Subsequently, we generated RsdA^{ΔC-terminal} and CsdA^{ΔN-terminal} mutants in *P. fici* to assess the functional significance of the protein interaction. Remarkably, both mutants displayed reduced growth rates, abnormal conidial morphology, and consistent regulation of certain secondary metabolites such as melanin and asperpentyn. This data is consistent with $\Delta csdA\Delta rsdA$ double mutant. These findings indicate that the interaction between CsdA and RsdA has a profound impact on the development and secondary metabolism of *P. fici*. These results have been incorporated into the fifth part of the manuscript's results section (see please Fig. 4).

Fig. 4. Mutations in the binding sites of CsdA and RsdA affect the development and secondary metabolism of *P. fici*. **a**, Measurement of radial growth in *P. fici* and its mutant strains ($n=5$). **b**, Comparative analysis of colonies and conidial morphology between *P. fici* and its mutant strains. **c**, HPLC analysis of SMs in *P. fici* and its mutant strains. UV absorptions at 254 nm are illustrated. **d**, Production of representative SMs in *P. fici* and its

mutant strains. All error bars are expressed as \pm SD. Statistical analysis was performed by using Two-way ANOVA ("ns": no significant. Significant at *** $p < 0.001$, **** $p < 0.0001$).

3. The same holds true for the RNA interaction. The authors demonstrate binding to the sequence GUCGGUAU. Here, a negative controls of similar RNA sequences is missing to demonstrate sequence specificity. To demonstrate that this interaction is functional meaningful, the CsdA binding site should be mutated in the background of the RsdA gene. According to the authors this should result in changes in alternative splicing and the resulting influence on development and secondary metabolism could be tested. Alternatively, constitutive expression of the alternatively splice version of *rsdA* might rescue the phenotype.

We have done the further experiments to address your concerns. To confirm the specificity of CsdA binding to the intron 2 sequence (GUCGGUAU) in *rsdA*, negative controls were synthesized *in vitro* using RNAs of intron 1 (1xGCAGGUCA) and intron 3 (1xGCCCGUAC). The results revealed that CsdA selectively binds to the GUCGGUAU sequence in intron 2, while showing no binding to intron 1 (GCAGGUCA) and intron 3 (GCCCGUAC). This indicates the specific binding of CsdA to the GUCGGUAU sequence in *rsdA* intron 2 (see please Fig. 5f). Furthermore, to validate the significance of the CsdA-*rsdA* pre-mRNA interaction, amino acid residues (R368-F373) within the binding site were mutated in *P. fici*. The splicing rate of *rsdA* intron 2 was found to increase by 1.6 times in the CsdA^{R368A-F373A} mutant, consistent with the splicing rate observed in the Δ *csdA* mutant (see please Fig. 5d). In addition, the CsdA^{R368A-F373A} mutant exhibited slow growth, abnormal conidial morphology, and alterations in secondary metabolites when compared to the control strain (Fig. 4). These observations further support the significance of the CsdA and *rsdA* pre-mRNA interaction in governing the growth, development, and secondary

metabolism of *P. fici*. These findings have been incorporated into the sixth part of the manuscript's results section (see please Fig. 4, 5d, f).

Fig. 5f. The binding of CsdA specificity to *rsdA* intron 2 was determined by EMSA assay. Nucleotides around the *rsdA* introns 5' splice sites were synthesized *in vitro* with GST-CsdA recombinant protein for EMSA assay.

Fig. 5d. The splicing efficiency of *rsdA* pre-mRNA in *CsdA*^{R368A-F373A} mutant was determined by qRT-PCR. All error bars are expressed as \pm SD. Statistical analysis was performed by using Two-way ANOVA ("ns": no significant. Significant at **** $p < 0.0001$).

Fig. 4a-c. Comparative analysis of radial growth and conidial morphology and SMS between the CsdA^{R368A-F373A} mutant and control strain (*P. fici* CK). All error bars are expressed as \pm SD. Statistical analysis was performed by using Two-way ANOVA (“ns”: no significant. Significant at **** $p < 0.0001$).

4. The role of CsdA in splicing needs to be tested transcriptome-wide. The RNA seq data of CsdA deletion strain are available and should be evaluated, for example but studying exon/exon spanning reads, that report the rate of splicing. It is very rare, that a RNA-binding protein has only a single target mRNA. In fact, the RNAseq experiments demonstrate that 1262 genes are differentially regulated in CsdA deletion mutants. 720 genes are co-regulated indicating that a substantial amount of mRNAs alter their expression independent of RsdA. Thus, one would expect more targets for such a global regulator.

We investigated the global impact of CsdA on alternative splicing in *P. fici* by analyzing transcriptome data. The distribution of exons and introns exhibited differences between the Δ *csdA* mutant and the control strain, with a significant reduction in intron retention rate observed in the Δ *csdA* mutant (Supplementary Fig. 15a). Moreover, by analyzing transcriptome data from Δ *csdA* mutant, we found about 180 exon skipping genes, which account for 1.2% of the total genes (Supplementary Fig. 15b) including several biosynthetic genes (Supplementary Fig. 15c). The GO enrichment analysis of these genes also showed that CsdA may have some other regulatory functions (Supplementary Fig. 15d). These results have been included in the discussion section (Supplementary Fig. 15).

Supplementary Figure 15. Global analysis of CsdA-mediated alternative splicing in *P. fici* by transcriptome data. **a**, Distribution of total introns and exons in the transcriptome of control strain and $\Delta csdA$ mutant in *P. fici*. **b**, Alternative splicing events in $\Delta csdA$ mutant compared with control strain in *P. fici*. **c**, Gene for exon skipping events in $\Delta csdA$ mutant. Orange represents genes that were significantly up-regulated and green represents genes that were significantly down-regulated ($|\log_2\text{foldchange}|>1$). **d**, GO enrichment analysis of exon skipping genes. All error bars are expressed as \pm SD. Statistical analysis was performed by using Two-way ANOVA.

Minor points:

1. Introduce RsdA in more detail. What type of global regulator is RsdA? A DNA-binding transcription factor?

We added the sentence to describe the function of RsdA in detail “A NCBI BLAST search of PfRsdA indicated the presence of a single predicted PAT1

domain at the N-terminus with an unclear function. However, despite the limited domain prediction, PfrsdA is involved in the regulation of over 50% of BGCs”.

2. In line 356, the lane number are not given correctly.

We have corrected the “lanes 1-3” in line 356 to “lanes 2-4”.

3. Line 370-373, what does $n=4$ indicates? If it is number of experiments, then the three replicates statement in the line 381 indicates the technical replicates? “ $n=5$ ” represents five biological replicates, and “Three replicates” in line 381 represent technical replicates. We have revised the sentence “Three replicates were performed for each analysis” in lines 381-382 to “Five biological replicates per strain ($n=5$)”.

4. In figure 4B, the arrow marks to indicate primers is not clear. It would be clearer if the authors could specify the reasons for different arrow colours (blue vs red) in the figure legend.

We added the sentence “The red dashes in the diagram represent the primers designed to amplify regions composed of exon-exon junctions after intron removal (splicing), while the blue dashes represent the primers designed within the introns that remain unspliced” in figure legend of the revised Fig. 5b.

5. In the model, Fig 6C, the domain architecture of RRM and ZnF rather seems to be disconnected into two parts.

We have made modifications to the model by drawing a connection line between the RRM and ZnF domains and replacing the original model accordingly.

6. In line 88, the argument to identify protein interaction partners of RsdA seems poor.

Thank you for the comments. The regulatory network of fungal secondary metabolism is highly intricate, with only a limited number of global regulators identified thus far, many of which exist in complex forms. LaeA serves as a well-studied example and can serve as a reference for exploring other regulatory complexes involved in fungal secondary metabolism. RsdA, a recently discovered regulator, lacks reported functional domains, suggesting that it may interact with other proteins to form complexes for the regulation of fungal secondary metabolism. Therefore, we revised the sentence as “In order to investigate the function of RsdA, we drew inspiration from the comprehensive regulatory role of the trimeric VeIB/VeA/LaeA complex and the KdmB-EcoA-RpdA-SntB chromatin complex in fungal development and secondary metabolism, as well as the presence of an unknown functional domain in RsdA. Based on these observations, we hypothesized that RsdA may engage in interactions with other proteins, potentially forming complexes that contribute to its regulatory function.”

7. While the text in line 148 suggests that in total of 2218 (39.8%) of metabolite ions were co-regulated by CsdA and RsdA, the correct number according to the following text and Fig. 2F is 1333.

The 2218 (39.8%) metabolic ions co-regulated by CsdA and RsdA in line 148 are correct, and the co-regulated metabolic ions in Fig. 2f have been corrected to 2218 (39.8%).

8. The authors have used purified, recombinant RsdA to pull down interaction partners from the total cell lysate. It is unclear to me why they didn't use the *in vivo* RsdA for the pull-down experiment.

That is an excellent question. In fact, we used *in vivo* RsdA for the pull-down experiment but perhaps did not emphasize this enough (lines 80-90). Given the prevalence of protein complexes in fungi, which can potentially lead to false positives and interfere with experimental results, we took measures to enhance the reliability of our experiments. To achieve this, we performed *in vitro* purification of a substantial quantity of recombinant RsdA protein. This allowed us to effectively capture the target protein from the cell lysate, thereby improving the accuracy and reliability of our experimental procedures.

9. In the methods section for "Transcriptional analysis by RNA-seq and qRT-PCR", the initial quality control steps used for RNA-seq read processing is missing. This section also misses the citation for methods such as HISAT and DESeq2 analysis.

The sentence "Data quality was controlled by fastp (version 0.19.7), and the sequencing error rate for a single base location was less than 1% (Goldstein, L.D. et al., PLoS One. (2016) 11(5): e0156132)." has been added to the methods section. Citations for the corresponding software, such as HISAT2 (Mortazavi, A. et al., Nat Methods. (2008) 5(7): 621-8) and DESeq2 (Love, M.I. et al., Genome Biol. (2014) 15(12): 550), are also added to the methods section.

10. The PCA analysis in FigED.1I showed high principal component variation between *csdA*Δ and *rsdA*Δ cells for the metabolite changes. If these proteins functions together in secondary metabolism, one might expect less variation as it has been observed for *A. fumigatus* in Fig 6B?

Thank you for the comments. We observed a distinct difference in the PCA analysis between the mutants of *P. fici* and *A. fumigatus*. This discrepancy can be attributed to the variations in metabolic ions and their abundances produced by the wild-type strains of *P. fici* and *A. fumigatus*. However, despite these differences, the overall regulatory trends of CsdA and RsdA remain consistent, highlighting the robustness of their regulatory roles across different fungal species. For example, DHN-melanin is increased 2-fold in $\Delta csdA$ mutant and 5.2-fold in $\Delta rsdA$ mutant, but it is negatively co-regulated by CsdA-RsdA (Fig. 6a). Moreover, the regulatory intensity of iso-A82775C in the two mutants were also different, but it is negatively co-regulated by CsdA-RsdA (Fig. 6c).

Reviewer 3:

The manuscript by Song et al. reports identification of the RNA-binding protein CsdA and posits that CsdA acts as a global regulator of fungal secondary metabolite production, in conjunction with another protein, the recently identified RsdA. In this current manuscript, the authors use pull-down in combination with other techniques to identify proteins that physically interact with RsdA. Among 10 candidates, one – CsdA – influences fungal development and small molecule production in a similar way as RsdA, and thus the authors then focus on CsdA. Very interestingly, it is then shown that RsdA and CsdA interact in the nucleus, and that CsdA may interact with the mRNA of RsdA (but see comment below).

Further it is shown that CsdA is widely conserved in filamentous fungi (usually with RsdA), and that deletion of either gene in diverse fungi strongly disrupts development, causes similar transcriptomic changes, and results in starkly altered metabolic profiles.

However, the principal claim of this paper – that CsdA (and RsdA) are regulators of global secondary metabolite production, is not supported by any of the presented data. CsdA (and RsdA) mutants have severe morphological defects and grow much slower, for example, it is demonstrated that both RsdA and CsdA play key roles in growth and conidial formation as well as pigmentation of *P. fici*. It is well established that almost any change in development affects secondary metabolite production. The strong growth, conidial, and pigmentation phenotypes observed indicate that development is seriously disrupted, and of course the spectrum of associated secondary metabolites will also be very different compared to WT. Similarly, many genes are changed in CsdA (and RsdA) mutants, and, unsurprisingly, among them are secondary metabolite gene clusters. Importantly, there's no evidence that either of the two genes specifically affects metabolism.

Thank you very much for your insightful comments. Based on your comments, we added evidence that CsdA (and RsdA) comprehensively regulate secondary metabolism, as well as relationship between development and metabolism. We have thoroughly addressed all the comments and suggestions provided, and we believe that the implemented improvements have significantly enhanced the content of the manuscript.

1. CsdA (and RsdA) are regulators of global secondary metabolite production.

Thank you for the suggestion. To clarify this, we added the additional data (refer to Supplementary Fig. 6) and rephrased the “global regulation” to “comprehensive regulation” in the appropriate text. Our analysis of transcriptome data revealed that 54 BGCs (71.05%) were significantly regulated in the $\Delta rsdA$ mutant, while 45 BGCs (59.21%) were significantly regulated in the $\Delta csdA$ mutant (refer to Supplementary Fig. 6). Furthermore, the analysis of metabolome data demonstrated that 64% of the metabolic ion products were significantly regulated in the $\Delta rsdA$ mutant, while 55% were significantly regulated in the $\Delta csdA$ mutant (see please Fig. 2e). These findings

indicate the substantial impact of RsdA and CsdA on the regulation of gene clusters and metabolic pathways associated with secondary metabolites. Considering the broad impact of their regulatory activities on secondary metabolites, we designate CsdA and RsdA as comprehensive regulators on SMs (Supplementary Fig. 6).

Supplementary Figure 6. Regulation of secondary metabolism by CsdA or RsdA alone. Significantly regulated BGC backbone genes were shown ($|\text{Log}_2\text{foldchange}| > 1$).

2. While I don't think the claim that CsdA is "a global regulator of secondary metabolism" makes sense, the work is very interesting – I wonder about the role of CsdA (and RsdA) in fungal development, and a broader exploration of the regulatory pathways controlling expression of CsdA (and RsdA) would likely reveal key insights in the control of fungal development. I recommend that the authors consider reframing their work, focusing on the role of CsdA (and RsdA) in overall development, which necessarily will also reveal how secondary metabolite production ties in.

Thank you for the insightful comments. We agree with you that fungal development is also very interesting in the regulation by this complex. Indeed,

several studies have connected secondary metabolism and fungal developmental process as reported previously (Calvo, A.M. et al., *Microbiol Mol Biol Rev.* (2002) 66(3): 447-59; Bayram, O. et al., *Science.* (2008) 320(5882):1504-6; Karahoda, B. et al., *Nucleic Acids Res.* (2022) 50(17):9797-9813). Our recent works have also reported that some BGCs or regulators linked fungal development to secondary metabolism, such as DHN-melanin, which is related to cell wall integrity (Zhang, P. et al., *Mol Microbiol.* (2017) 105(3): 469-483), and CsnE, a regulator that coordinates fungal SM and conidial formation (Zheng, Y. et al., *Sci China Life Sci.* (2017) 60(6): 656-664). In this work, we mainly focused on the regulation of CsdA and RsdA on secondary metabolism and presented the solid data to support the conclusions. We appreciate your suggestions and are going to do the deep study how CsdA/RsdA complex regulation on fungal developments and its biological significance.

Minor points:

- homology of CsdA (in fungi and across eukaryotes) must be discussed more clearly, including the finding that the fungal RRM includes the ribonucleoprotein domains 1 and 2 (in line 231 is says, incorrectly, “ribonucleoproteins 1 and 2 (RNP1, RNP2)”).

In the discussion section (line 327), we added the description of homology of CsdA in fungi and eukaryotes as follows: “CsdA homologs show high degrees of amino acid identity in Ascomycota (more than 40%), but lower identity in other eukaryotes due to the relatively small proportion of functional domains in CsdA (Supplementary Fig. 1h), e.g., 24%/46% identity and 22%/6% coverage in plants (*Oryza sativa*) /animals (*Homo sapiens*), respectively”.

In the results section (line 231), we revised the sentence as “By multiple alignments of RRM and ZnF from *P. fici*, *A. fumigatus*, and *A. nidulans* with

sequences reported in humans, we identified conserved ribonucleoprotein domains 1 and 2 (RNP1, RNP2) in the fungal RRM domain and a conserved motif (W-X-C-X₂₋₄-C-X₃-N-X₆-C-X₂-C) belonging to the “CCCC” type of ZnF (Supplementary Fig. 11a-d”).

- the hypothesis that CsdA binds to the pre-mRNA of *rsdA* is exciting, but there seems to be a lack of controls in the experiments described in the section starting in line 245. I understand that docking suggested that CsdA binds to the GUCGGUAU motif, but for the following electrophoretic mobility shift assay results to be meaningful, I think it would be necessary to include other control sequences, ideally from the same pre-mRNA. Moreover, the binding affinities don't seem very high (micromolar) – here it would help to give some perspective, e.g., by comparing these values with binding data for other protein-mRNA complexes.

We have done the further experiments to address your concerns. To confirm the specificity of CsdA binding to the intron 2 sequence (GUCGGUAU) in *rsdA*, negative controls were synthesized *in vitro* using RNAs of intron 1 (1xGCAGGUCA) and intron 3 (1xGCCCGUAC). The results revealed that CsdA selectively binds to the GUCGGUAU sequence in intron 2, while showing no binding to intron 1 (GCAGGUCA) and intron 3 (GCCCGUAC). This indicates the specific binding of CsdA to the GUCGGUAU sequence in *rsdA* intron 2 (see please Fig. 5f).

Based on reported binding data for protein-RNA complexes, for example, *S. pombe* U2AF23 complex binds to 5'-CUAGG-3' RNA with K_d value of 1.2 μ M (Yoshida, H. et al., *Genes Dev.* (2015) 29(15): 1649-60), Tra2- β 1 binds to 5'-AAGAAC-3' RNA with K_d value of 2.25 μ M (Cléry, A. et al., *Nat Struct Mol Biol.* (2011) 18: 443-450), and hnRNPG binds to 5'-AUCAAA-3' RNA with K_d value of 18 μ M (Moursy, A. et al., *Nucleic Acids Res.* (2014) 42(10): 6659-72), suggested that the K_d value of CsdA with 5'-GUCGGUAU-3' was reliable.

- the introduction is overly long and not focused. For example, lines 38-40 contain trivial statements about basic ecology that seem unnecessary.

We have removed the sentence “In nature, fungi have been encompassed by a variety of biotic and abiotic factors over long evolutionary periods. In order to adapt to various complex environments, fungal BGCs are coordinated to produce specific active small molecules.” from the manuscript and carefully revised the entire introduction to make it more concise and readable.

Reviewer 4:

“Fungal secondary metabolism is governed by an RNA-binding protein CsdA/RsdA complex” by Song et al.

Review Summary

Fungal secondary metabolites (SMs) have been rich sources of drug discovery, as exemplified by the historic drugs penicillin and lovastatin. However, a large portion of fungal SMs are still hidden, due to highly context-specific regulation of biosynthetic gene clusters (BGCs). In this manuscript, Song et al. tries to tackle this limitation by investigating the molecular action mechanisms of global regulator of SM, RsdA, which accounts for the regulation of more than 50% of BGCs throughout the filamentous fungi species. The authors identified CsdA as a novel interactor of RsdA protein, and as an RNA-binding protein regulating alternative splicing of *rsdA* pre-mRNAs as well. Comprehensive transcriptomics and metabolomics analysis revealed that RsdA and CsdA are functionally intertwined to co-regulate a large portion of secondary metabolites and BGCs

(39.8 % of metabolic ions and about 24 key BGC backbone genes out of 76 genes).

Overall, this manuscript provides novel mechanistic insights for the global regulation of fungal SMs which may open up new possibilities of discovering many hidden SMs that might be useful for new drug/chemical screenings. Therefore, I believe the study is of interest to broad audiences from microbiology, RNA biology, drug discovery and agriculture. However, at the same time, the conclusions are often based only on weak experimental evidence or from simple speculations, thus not yet strong enough, especially for being published in Nature communications. I would suggest 3 major points and a few minor suggestions for the authors to revisit to make the manuscript more conclusive and useful to wide audiences.

Thank you for the insightful comments. According to your suggestions, we added more data to support our conclusion, and rearranged the revised manuscript to make it more conclusive. We believe that the quality of manuscripts has been greatly improved. Below, we have detailed responses to all the comments.

Major points:

1. The functional significance of CsdA-RsdA protein complex:

The authors identified CsdA as a new interactor of RsdA and proposed it as a regulator of RsdA in two folds: RsdA protein function and *rsdA* pre-mRNA splicing. Unlike the pre-mRNA part, the functional significance of CsdA-RsdA protein complex in coordinating secondary metabolism is not clearly demonstrated, even though the authors show direct protein-protein interaction and nuclear co-localization between CsdA and RsdA. Likewise, the functional significance of CsdA-RsdA protein interaction in either the CsdA's role as an RBP or the RsdA's role as a global TF regulator are not thoroughly investigated.

Therefore, it is still unclear to me whether the CsdA-RsdA protein complex has any functional roles in secondary metabolism or BGC regulation. I would suggest for the authors to perform some more experiments to address these concerns. Below are some questions and suggestions related to the specific point, but the authors can choose any other experiments to ultimately demonstrate the functional significance of CsdA/RsdA protein complex in secondary metabolism and BGC regulation.

- What are the changes of SMs (or BGC changes) in delta-csdA/delta-rsdA double mutant as compared with single delta-rsdA or single delta-csdA mutant?

The BGC backbone gene is crucial for SM synthesis and plays an essential role in the alteration of SMs and thus can be used as a proxy for BGC regulation. We examined the regulated BGC backbone genes in the double mutant $\Delta AfcsdA\Delta AfrsdA$ compared to the single mutants $\Delta AfcsdA$ or $\Delta AfrsdA$ in *A. fumigatus*. The findings revealed that 21 BGCs (53.85%) exhibited significant regulation in $\Delta AfrsdA$, 22 BGCs (56.41%) in $\Delta AfcsdA$, and 20 BGCs (51.28%) in $\Delta AfrsdA\Delta AfcsdA$. Among them, 10 BGCs displayed more pronounced regulatory effects in the double mutant than in the single mutants.

- Can the CsdA point mutant which is deficient in RsdA interaction be generated by considering the structural modeling in Fig3d? For example, by mutating H26 and/or Y29 to A and/or F? If so, then can this mutant still be able to rescue delta-csdA phenotypes?

According to the structural model of Fig. 3d and *in vivo* pull-down assays, the C-terminal of RsdA and the N-terminal of CsdA play crucial roles in these interactions. Firstly, we generated RsdA^{ΔC-terminal} and CsdA^{ΔN-terminal} mutants in *P. fici* to assess the functional significance of the protein interaction. Remarkably, both mutants displayed reduced growth rates, abnormal conidial morphology, and consistent regulation of certain secondary metabolites such as melanin and

asperpentyn. Secondly, as you suggested, we mutated amino acid residues (H26-Y29) at the CsdA-RsdA binding site, and the CsdA^{H26A-Y29A} mutant's phenotype is consistent with that of the Δ csdA mutant. These findings indicate that the interaction between CsdA and RsdA has a profound impact on the development and secondary metabolism of *P. fici*. Reviewer 2's major point 2 also addresses the findings mentioned in these results and have been incorporated into the fourth part of the manuscript's results section (see please Fig. 4).

- Can the delta-csdA phenotypes be rescued by the complementation of rsdA mature mRNA? If rsdA mature mRNA can fully rescue delta-csdA phenotypes, then one may conclude that CsdA's role in secondary metabolism and fungi growth is mainly via alternative splicing of rsdA pre-mRNA. On the other hand, if this is not the case, CsdA must have additional roles in modulating RsdA protein function or RsdA-independent roles in the secondary metabolism.

We complemented ORF of *rsdA* in a Δ AfcsdA Δ AfrsdA mutant of *A. fumigatus*. The results showed that the colony diameter and conidial number of the mutant were consistent with those of the Δ AfcsdA Δ AfrsdA mutant, indicating that the complement of *rsdA* mature mRNA could not completely rescue the morphological phenotypes of the Δ AfcsdA mutant. In contrast, the metabolic profile of the Δ AfcsdA Δ AfrsdA^{rsdA com} mutant was consistent with the control strain of *A. fumigatus*, suggesting that complement *rsdA* mature mRNA could rescue the metabolic changes in Δ AfcsdA mutant to a certain extent. These results support that the role of CsdA in secondary metabolism is mainly *via* alternative splicing of *rsdA* pre-mRNA.

- For CsdA to function as a splicing factor, it should first be in the nucleus but CsdA has no (putative) NLS. Is the nuclear localization of CsdA dependent on

its interaction to RsdA? This can simply be tested by analyzing CsdA localization in delta-rsdA background.

RNA binding proteins are usually expressed and function in the nucleus (Han, J. et al., Trends Cell Biol. (2011) 21(6): 336-43). According to your suggestion, we deleted *AfrsdA* in the AfCsdA-mCherry strain of *A. fumigatus*. By fluorescence detection of the mutant, we found that CsdA was still located in the nucleus and co-located with the DAPI stained nucleus (Please see Figure below).

Figure. The nuclear localization of AfCsdA in $\Delta AfrsdA$ mutant of *A. fumigatus*. CsdA-mCherry localizes on the nucleus and co-localize with DAPI signals. BF, bright-field. DAPI: 4', 6-diamidino-2-phenylindole. Scale bars, 5 μm .

2. What are the conditions (e.g., environmental stresses or upstream signalings) in which CsdA-RsdA expressions/activities are regulated (increased or decreased) for fungal growth and SMs generation? For example, can LaeA activate CsdA-RsdA?

Thanks. This is a big question. Fungal growth and SMs are controlled by a very complex hierarchical regulatory network. As your suggestions, we conducted an intensive analysis of transcriptome data of CsdA (and RsdA). First, we analyzed the expression of *AfcsdA* and *AfrsdA* in a $\Delta AflaeA$ mutant of *A. fumigatus*, and found no significant change in its expression compared with the control strain of *A. fumigatus* ($\text{Log}_2\text{foldchange} < 1$). Furthermore, GO enrichment analysis of 720 genes significantly co-regulated by CsdA and RsdA showed that

they were mainly related to fungal response to oxidative stress. Therefore, we believe that oxidative stress in the environment is the possible conditions for CsdA-RsdA expression/activation.

3. What will be the potential applications of the findings (e.g., specific inhibitors or activators acting on CsdA-RsdA interface for inducing novel SMs)? Can the authors at least perform some proof-of-concept experiments for possible applications?

Thank you for the suggestions. I believe that our findings have the potential applications in the discovery of new natural products and other aspects. In our previous studies, we successfully identified 15 novel polyketides through the manipulation of the epigenetic regulators *cclA* and *hdaA* in *P. fici* (Wu, G. et al., *Org Lett.* (2016) 18(8):1832-5). Building upon this knowledge, we propose three promising avenues for further exploration.

Firstly, CsdA, functioning as a comprehensive regulator of fungal secondary metabolism, also acts as an RNA binding protein with splicing capabilities. By genetically modifying CsdA to generate transcripts that can be translated into diverse functional proteins through intron inclusion or removal, we can potentially activate the production of novel secondary metabolites. This approach holds great promise for expanding the repertoire of bioactive compounds.

Secondly, our findings shed light on the complex regulatory network of fungi. While current understanding of gene regulation predominantly focuses on epigenetic and transcriptional mechanisms, the role of post-transcriptional regulation in fungal secondary metabolism remains largely unexplored. The regulatory patterns we have uncovered offer valuable insights into this understudied aspect, contributing to a more comprehensive understanding of fungal gene regulation.

Lastly, eukaryotic alternative splicing is governed by spliceosomes, yet the assembly and regulatory mechanisms of spliceosomes in fungi are not fully elucidated. The RBP complex discovered in our study presents a valuable opportunity for investigating the structural biology of fungi, particularly regarding spliceosome assembly and regulation.

By pursuing these directions, we can further unravel the intricate mechanisms underlying fungal secondary metabolism and enhance our knowledge of fungal gene regulation as a whole. We will perform those studies to test our ideas in the near future.

Other minor points:

1. It is not yet clearly demonstrated that the RsdA-CsdA protein complex itself plays important roles in fungal secondary metabolism. The manuscript title and other texts in the abstract and discussion should be toned down unless this would be confirmed by further experimentations. The model in figure 6c should also be corrected.

In major point 1, we have demonstrated the role of CsdA-RsdA complex in fungal secondary metabolism by mutation at binding sites. Thus, the title and other texts of the manuscript, as well as revised Fig. 8, were confirmed.

2. Throughout Fig1 and 2, delta-csdA and delta-rsdA show similar but different phenotypes in terms of growth and secondary metabolites. Can these phenotypes be explained by DEGs from RNA-seq?

The DEGs provide information that can inform the differences in growth and secondary metabolites. For example, by analyzing the DEGs associated with fungal growth and asexual development, we found that *fadA* and *vosA* were significantly regulated in $\Delta csdA$ and $\Delta rsdA$ mutants, with *fadA* in opposite mode

and *vosA* in the same mode in both mutants (see please Supplementary Fig. 5). Thus, we believe that these differentially expressed genes could be related to the regulation of CsdA and RsdA on the development of *P. fici*. In the regulation of secondary metabolites, 54 BGCs (71.05%) were significantly regulated in Δ *rsdA* mutant and 45 BGCs (59.21%) in Δ *csdA* mutant (see please Supplementary Fig. 6). In addition to 24 BGCs co-regulated by CsdA and RsdA, there were 29 BGCs separately regulated by RsdA and 21 BGCs separately regulated by CsdA, such as *pta* cluster (see please Fig. 2g). These results suggest that these different phenotypes may be explained by DEGs from RNA-seq.

It would be nicer to show whether the co-regulated BGC backbone genes in Fig2g are more related to the coregulated metabolites in Fig 2e-f. More generally, do the BGC backbone genes regulated by CsdA and/or RsdA show correlation with the down or upregulated metabolic ions by CsdA and/or RsdA shown in Fig 2e-f?

Fungal SMs are synthesized by BGCs, so the expression of BGC backbone genes usually has a decisive influence on the ion abundance of SMs. So far, only three BGCs have been reported to be associated with SMs in *P. fici*. As shown in revised Fig. 6, in the two mutants, significant up-regulation of *pfma* cluster backbone gene resulted in increased melanin production, while significant down-regulation of *iac* cluster backbone gene resulted in significant decrease in the metabolic ion abundance of iso-A82775C. However, the backbone genes in *pta* cluster were significantly up-regulated in Δ *csdA* mutant and down-regulated in Δ *rsdA* mutant. These results suggest a correlation between BGC backbone gene expression in Fig. 2g and metabolic ion abundance in Fig. 2e-f.

Do the BGC genes (or other DEGs) that are oppositely regulated by CsdA and RsdA regulate conidial formation and may explain the differential phenotypes?

Fungal development and secondary metabolism are usually a unified process (Calvo, A.M. et al., *Microbiol Mol Biol Rev.* (2002) 66(3): 447-59). We took *PFICL_15056*, a BGC backbone gene in *P. fici* that is inversely regulated by CsdA and RsdA, as an example to analyze the correlation between differentially regulated BGC and conidial formation. We found that deletion of *PFICL_15056* did not affect conidial formation. These results suggest that the differential phenotypes in different mutants cannot be explained by BGC genes. As in the first question in minor point 2, we have shown that these differential phenotypes can be explained by DEGs from RNA-seq. We want to emphasize that both sporulation and secondary metabolism are complex processes in fungi and can not always be explained by BGC genes or other DEGs.

3. How the alternative splicing of *rsdA* affects *rsdA* mature mRNA level? Would intron2 inclusion increase *rsdA* mRNA stability? What would be the consequences of intron2 inclusion or exclusion to the final RsdA protein sequence, if any?

First, we measured *rsdA* mRNA stability in Δ *csdA* mutant of *P. fici*, and the results showed that the half-life of *rsdA* mRNA was shortened compared with the control strain, indicating that the deletion of *csdA* reduced the stability of *rsdA* mRNA and resulted in the reduction of *rsdA* mature mRNA level (see Figure below). Second, intron 2 inclusion may affect *rsdA* mRNA stability, because the control strain had more *rsdA* intron 2 retention and higher mRNA stability compared with the Δ *csdA* mutant. Finally, if RsdA protein contains intron 2, its translation will be terminated prematurely, affecting the normal development and metabolism of *P. fici*. If intron 2 is removed, it will be translated to form the correct protein and maintain the normal development and metabolism of *P. fici*.

Figure. Stability of *rsdA* mRNA in $\Delta csdA$ mutant compared with control strain (*P. fici* CK).

$t_{1/2}$: the half-life of mRNA. ActD: actinomycin D.

4. Would it be possible to analyze global splicing in a more unbiased and quantitative way from RNA-seq data instead of simply cherry-picking some genes of interests (as in Fig4 and 5)? The authors may find more target genes that are alternatively spliced by CsdA and regulate secondary metabolism.

We investigated the global impact of CsdA on alternative splicing in *P. fici* by analyzing transcriptome data. The distribution of exons and introns exhibited differences between the $\Delta csdA$ mutant and the control strain, with a significant reduction in intron retention rate observed in the $\Delta csdA$ mutant (Supplementary Fig. 15a). Moreover, by analyzing transcriptome data from $\Delta csdA$ mutant, we found about 180 exon skipping genes, which account for 1.2% of the total genes (Supplementary Fig. 15b) including several biosynthetic genes (Supplementary Fig. 15c). The GO enrichment analysis of these genes also showed that CsdA may have some other regulatory functions (Supplementary Fig. 15d). We also answered the same question in reviewer 2's major point 4 and added the results to the discussion section (Supplementary Fig. 15).

5. Where are the binding sites of CsdA (GUCGGUAU) located in the *rsdA* gene? Are there single or many GUCGGUAU motifs and are they enriched near exon2-intron2 junction, intron2-exon3 junction or other sites? What would be

the possible molecular mechanisms of CsdA in regulating splicing in terms of its possible interactions with the components of spliceosome?

First, by searching GUCGGUAAU on *rsdA* pre-mRNA, we found a single motif located at the exon2-intron2 junction. Second, the splicing process in eukaryotes is usually regulated by the spliceosome, so we believe that CsdA is related to the components of the spliceosome in the regulation of splicing. According to Supplementary Fig. 11, ZnF domains (PfZnF1, PfZnF2) in CsdA are responsible for the recognition of pre-mRNA, so we obtained the homologous protein ZRANB2 of PfZnF domains. ZRANB2 is an SR-like nuclear protein that regulates the splicing process by interacting with the spliceosomal proteins U1-70K and U2AF35 (Mangs, A.H. & Morris, B.J., *Int J Biochem Cell Biol.* (2008) 40(11):2353-7). Since both PfZnF and ZRANB2 have conserved motifs (W-X-C-X₂₋₄-C-X₃-N-X₆-C-X₂-C) and belong to "CCCC" type ZnFs, CsdA shows the possibility of interacting with spliceosome components to regulate the splice process.

REVIEWER COMMENTS

Reviewer #2 (Remarks to the Author):

This is a revised version of the manuscript by Song et al. describing the role of the CsdA/RsdA complex in the endophyte *P. fici*. The authors have advanced the study by introducing new interesting mutants that affect the interaction between the two proteins and inhibit the RNA binding activity of CsdA.

Unfortunately, they have not fully utilized the potential of these mutants to dissect the individual function of CsdA, RsdA, and the CsdA/RsdA complex.

As a result, the precise function of the complex remains unclear and therefore the current manuscript does not merit publication in Nature Communications. Major points of criticism are still valid:

1. It is crucial to dissect the functions of these important regulators. What are the specific roles of CsdA, RsdA, and the complex they form? This could be addressed by analyzing mutants that are unable to interact, preferably through point mutations that completely disrupt the complex. Such mutants would allow the identification of targets that rely on the complex. Unfortunately, the results regarding the intriguing mutant H26-Y29 are only briefly presented.

2. The authors need to further investigate the role of the regulation at the level of alternative splicing. Deletion of CsdA leads to decreased mRNA levels of *rsdA*, and the splicing pattern of intron 2 is altered in the absence of CsdA. To gain a deeper understanding of this type of regulation, the authors should devise a compelling strategy to investigate it thoroughly.

Reviewer #3 (Remarks to the Author):

The revised manuscript by Song et al. includes impressive additional data, most importantly demonstrating that the C-terminal and N-terminal domains of RsdA and CsdA, respectively, are required for their interactions, and demonstrating that CsdA binding to the intron 2 sequence (GUCGGUAU) in *rsdA* is specific, by testing additional negative controls. In addition, the data for the overall impact of CsdA on splicing help better define the role of this RBP.

The manuscript has been improved in many other ways, and the authors now use more cautious wording when describing the impact of RsdA and CsdA on fungal secondary metabolite production. While substituting “globally” with “comprehensively” makes sense in many instances, in some cases “global(ly)” may be entirely appropriate. My main concern (and I believe also of some of the other reviewers) was to single out secondary metabolite production as the one aspect of fungal physiology that would be regulated by either gene; instead, it is “fungal development AND secondary metabolite production” that is globally/comprehensively regulated, as would be expected since development and secondary metabolite production are intricately linked. Thus statements such as “RsdA is a global/comprehensive regulator of fungal metabolism” (e.g. line 367/8 and 393) are not quite correct and, to a non-expert, would be misleading. Therefore, the authors should consistently adopt the wording in lines 382/3: “... CsdA regulates fungal development and metabolism ...”.

Minor points (line numbers refer to the pdf with tracked changes):

line 249: change to “dramatically changed”

line 280: something is missing here, e.g., “using HADDOCK”.

line 295: “Calorimetry”

line 314: “indicated” seems too strong; I’s suggest “suggested that CsdA may regulate” instead.

line 461: (“ns”: not significant).

line 476/7: “The binding of CsdA 476 specificity to rsdA intron 2 was determined”. Revise, e.g. to “The specificity of CsdA binding to rsdA intron 2 was determined ...”

line 498: “functional consistency of RNA-binding protein CsdA/RsdA complex in eukaryotic metabolism” – this is not supported by the data. The authors looked at *A. fumigatus*, demonstrating functional conservation in one branch of fungi. Similar statements in the main text should also be amended.

line 500: “The metabolome was used to determine the comprehensive metabolites regulated by CsdA and RsdA in *A. fumigatus*.” Revise to something like: “Comparative metabolomics of WT and CsdA and RsdA knockout strains was used to ...”.

line 504: the legend to Figure 8 needs to be revised to correct grammar/syntax issues. .

Reviewer #4 (Remarks to the Author):

In the revised manuscript, the authors addressed almost all my questions and concerns with extensive experiments and discussion. However, I still have a remaining concern about the final model regarding the posttranscriptional *rsdA* mRNA regulation by CsdA. According to the authors model, CsdA promotes *rsdA* expression by inhibiting intron 2 splicing and producing more RsdA protein (line 385-386). However, for normal RsdA protein expression, intron 2 should be spliced out from *rsdA* mRNA before its nuclear export for functional RsdA protein expression (translation) in the cytosol, otherwise it would generate truncated RsdA protein (according to their explanation in the response letter) or the unspliced RNAs would be rapidly degraded via Nonsense Mediated mRNA Decay (NMD) pathway. Therefore, I rather feel that the normal function of CsdA is to delay the maturation of *rsdA* mRNA by inhibiting intron 2 splicing. And perhaps, this may help to keep the functional RsdA protein level in the optimal range.

Related to this, is the reduced *rsdA* RNA expression in delta-*csdA* (Fig 5a) transcriptional or posttranscriptional? If it is transcriptional, the authors should directly demonstrate it by experiments such as Pol II CHIP-seq, etc. If it is posttranscriptional, I feel the rationale is still missing regarding the molecular mechanisms how the increased splicing of intron 2 results in the destabilization of the mRNA. One possible explanation might be that the *rsdA* level measured in Fig 5a might have captured both pre- and mature forms of *rsdA* mRNA and mature *rsdA* mRNA has much shorter half-life than pre-mRNA. If so, the higher expression of *rsdA* in the wildtype is due to the higher proportion of pre-mRNA than mature mRNA.

That being said, I still feel that the revised manuscript has been much improved to merit its publication

in Nature communications once the authors provide at least some plausible explanations about my questions above.

Reviewer #2 (Remarks to the Author):

This is a revised version of the manuscript by Song et al. describing the role of the CsdA/RsdA complex in the endophyte *P. fici*. The authors have advanced the study by introducing new interesting mutants that affect the interaction between the two proteins and inhibit the RNA binding activity of CsdA. Unfortunately, they have not fully utilized the potential of these mutants to dissect the individual function of CsdA, RsdA, and the CsdA/RsdA complex.

As a result, the precise function of the complex remains unclear and therefore the current manuscript does not merit publication in Nature Communications.

Thanks again for the positive comments and useful suggestions. In the revised version, we have precisely analyzed the individual function of CsdA, RsdA, and CsdA/RsdA complex and rearranged the relevant text. See detailed responses below.

Major points of criticism are still valid:

1. It is crucial to dissect the functions of these important regulators. What are the specific roles of CsdA, RsdA, and the complex they form? This could be addressed by analyzing mutants that are unable to interact, preferably through point mutations that completely disrupt the complex. Such mutants would allow the identification of targets that rely on the complex. Unfortunately, the results regarding the intriguing mutant H26-Y29 are only briefly presented.

To address the individual function of CsdA, RsdA and complex, we have done functional enrichment analysis and metabolic analysis from transcriptomic and metabolomic levels. From the transcriptomic level, the metabolic pathway genes involved in CsdA, RsdA or CsdA-RsdA (co-regulated 720 genes) are mainly carbon metabolism, steroid biosynthesis and ABC transporters. Specifically, CsdA, RsdA, or CsdA-RsdA regulates some unique metabolic pathways, respectively, such as CsdA regulation on TCA cycle, RsdA regulation

on amino acid metabolism, CsdA-RsdA regulation on terpenoid biosynthesis (Supplementary Fig. 11a). These results suggest that the functions of CsdA are involved in carbon metabolism and TCA cycle, RsdA are involved in carbon metabolism and amino acid metabolism, and CsdA-RsdA complex are involved in carbon metabolism and terpenoid biosynthesis in the growth, development and metabolism of *P. fici*.

From the metabolomic level, we thoroughly analyzed the function of complex by comparing the metabolome data of CsdA^{H26A-Y29A} and CsdA^{ΔN-terminal} mutants with control strain. In consistence with $\Delta csdA$ or $\Delta rsdA$ mutant, the metabolic ion peaks of 2830 (39%) and 3923 (55%) were significantly affected in the mutation of the binding sites H26-Y29 in the complex or complete deletion of the interaction region (*N*-terminus of CsdA), respectively (Supplementary Fig. 11b). In addition, 1977 (28%) of the differentially regulated metabolic ion peaks were co-regulated in CsdA^{H26A-Y29A} and CsdA^{ΔN-terminal} mutants (e.g., iso-A82775C) compared with $\Delta csdA$ or $\Delta rsdA$ mutant, suggesting that this complex has an individual function in metabolite regulation (Supplementary Fig. 11c). Moreover, CsdA or RsdA regulates individually the metabolic ion peaks of 255 (4%) or 1510 (21%) (e.g., pestheic acid), respectively. These results have been incorporated into the fourth part in the manuscript's results section (see please Supplementary Fig. 11).

Supplementary Figure 11. The individual functions of CsdA and RsdA and their complex were evaluated by DEGs and metabolome data analysis. a. KEGG was used to analyze the metabolic pathways of CsdA and RsdA respectively and their co-regulated genes. The red fonts represent metabolic pathways involved in CsdA and RsdA and the complex. Cyan represents metabolic pathways individually involved in the complex. Carbon metabolism pathway is the main metabolic pathway involved in all three. **b.** Volcano plots show the differentially regulated metabolic ion peaks of CsdA^{H26A-Y29A} or CsdA^{ΔN-terminal} mutant versus the control. **c.** Distribution of total metabolic ion peaks due to mutations in CsdA protein interaction sites or regions in *P. fici*.

2. The authors need to further investigate the role of the regulation at the level of alternative splicing. Deletion of CsdA leads to decreased mRNA levels of *rsdA*, and the splicing pattern of intron 2 is altered in the absence of CsdA. To gain a deeper understanding of this type of regulation, the authors should devise a compelling strategy to investigate it thoroughly.

As you know, eukaryotic DNA to mature RNA requires two steps, first, DNA to pre-mRNA containing introns at the transcriptional level, and then pre-mRNA to mature mRNA with introns removed at the post-transcriptional level. To verify that CsdA affects the stability of *rsdA* intron 2 splicing at the post-transcriptional level and leads to a decrease in the abundance of mature mRNA, we measured the abundance of intron 2 in Δ *csdA* mutant at the transcriptional (pre-mRNA) and post-transcriptional levels (mRNA), respectively. The results showed that the abundance of *rsdA* intron 2 in Δ *csdA* mutant was not significantly different at the transcriptional level compared with the control, but was significantly decreased at the post-transcriptional level by more than 2 times, indicating that the regulatory effect of CsdA is likely post-transcriptional rather than transcriptional levels. Therefore, these results support our hypothesis that CsdA affects the stability of *rsdA* intron 2 splicing at the post-transcriptional level, leading to a decrease in the abundance of mature mRNA. These results have been incorporated into the fifth part in the manuscript's results section (see please Supplementary Figure 13).

Supplementary Figure 13. Relative abundance of *rsdA* intron 2 at transcriptional and post-transcriptional levels in Δ *csdA* mutant compared with the control. “pre-mRNA” represents

the transcriptional level and “mRNA” represents the post-transcriptional level. The red dashes in the diagram represent the primers designed to amplify regions composed of exon-exon junctions after intron removal (mRNA), while the blue dashes represent the primers designed within the introns that remain unspliced (pre-mRNA). All error bars are expressed as \pm SD. Statistical analysis was performed by using Two-way ANOVA (“ns”: not significant. Significant at $*p < 0.05$).

Reviewer #3 (Remarks to the Author)

The revised manuscript by Song et al. includes impressive additional data, most importantly demonstrating that the C-terminal and N-terminal domains of RsdA and CsdA, respectively, are required for their interactions, and demonstrating that CsdA binding to the intron 2 sequence (GUCGGUAU) in *rsdA* is specific, by testing additional negative controls. In addition, the data for the overall impact of CsdA on splicing help better define the role of this RBP.

The manuscript has been improved in many other ways, and the authors now use more cautious wording when describing the impact of RsdA and CsdA on fungal secondary metabolite production. While substituting “globally” with “comprehensively” makes sense in many instances, in some cases “global(ly)” may be entirely appropriate. My main concern (and I believe also of some of the other reviewers) was to single out secondary metabolite production as the one aspect of fungal physiology that would be regulated by either gene; instead, it is “fungal development and secondary metabolite production” that is globally/comprehensively regulated, as would be expected since development and secondary metabolite production are intricately linked. Thus statements such as “RsdA is a global/comprehensive regulator of fungal metabolism” (e.g. line 367/8 and 393) are not quite correct and, to a non-expert, would be misleading. Therefore, the authors should consistently adopt the wording in lines 382/3: “... CsdA regulates fungal development and metabolism ...”.

Thank you very much for the positive comments. According to your suggestions, we have corrected all similar wording in the revision manuscript. We believe that the quality of manuscript has been greatly improved.

Minor points (line numbers refer to the pdf with tracked changes):

line 249: change to “dramatically changed”.

Done.

line 280: something is missing here, e.g., “using HADDOCK”.

We revised the sentence as “...and performed molecular docking with the CsdA protein *via* the HADDOCK website”.

line 295: “Calorimetry”.

“Isothermal Titration Calorimeter” was corrected to “Isothermal Titration Calorimetry”.

line 314: “indicated” seems too strong; I’s suggest “suggested that CsdA may regulate” instead.

We revised the sentence as “This suggested that CsdA may regulate DHN-melanin synthesis through RsdA”.

line 461: (“ns”: not significant).

Done.

line 476/7: “The binding of CsdA specificity to *rsdA* intron 2 was determined”.
Revise, e.g. to “The specificity of CsdA binding to *rsdA* intron 2 was determined ...”.

We revised the sentence as “The specificity of CsdA binding to *rsdA* intron 2 was determined by EMSA assay”.

line 498: “functional consistency of RNA-binding protein CsdA/RsdA complex in eukaryotic metabolism” – this is not supported by the data. The authors looked at *A. fumigatus*, demonstrating functional conservation in one branch of fungi. Similar statements in the main text should also be amended.

The “eukaryotic metabolism” was corrected to “fungal metabolism”, and similar problems were corrected in the entire texts.

line 500: “The metabolome was used to determine the comprehensive metabolites regulated by CsdA and RsdA in *A. fumigatus*.” Revise to something like: “Comparative metabolomics of WT and CsdA and RsdA knockout strains was used to ...”.

We revised the sentence as “Comparative metabolomics of WT and *csdA* and *rsdA* knockout strains was used to determine the metabolic changes in *A. fumigatus*”.

line 504: the legend to Figure 8 needs to be revised to correct grammar/syntax issues.

We revised the sentence as “The model shows that CsdA/RsdA interaction with each other to coordinate fungal secondary metabolism and development”.

Reviewer #4 (Remarks to the Author)

In the revised manuscript, the authors addressed almost all my questions and concerns with extensive experiments and discussion. However, I still have a remaining concern about the final model regarding the posttranscriptional *rsdA* mRNA regulation by CsdA. According to the authors model, CsdA promotes *rsdA* expression by inhibiting intron 2 splicing and producing more RsdA protein (line 385-386). However, for normal RsdA protein expression, intron 2 should be spliced out from *rsdA* mRNA before its nuclear export for functional RsdA protein expression (translation) in the cytosol, otherwise it would generate truncated RsdA protein (according to their explanation in the response letter) or the unspliced RNAs would be rapidly degraded via Nonsense Mediated mRNA Decay (NMD) pathway. Therefore, I rather feel that the normal function of CsdA is to delay the maturation of *rsdA* mRNA by inhibiting intron 2 splicing. And perhaps, this may help to keep the functional RsdA protein level in the optimal range.

Related to this, is the reduced *rsdA* RNA expression in delta-*csdA* (Fig 5a) transcriptional or posttranscriptional? If it is transcriptional, the authors should directly demonstrate it by experiments such as Pol II CHIP-seq, etc. If it is posttranscriptional, I feel the rationale is still missing regarding the molecular mechanisms how the increased splicing of intron 2 results in the destabilization of the mRNA. One possible explanation might be that the *rsdA* level measured in Fig 5a might have captured both pre- and mature forms of *rsdA* mRNA and mature *rsdA* mRNA has much shorter half-life than pre-mRNA. If so, the higher expression of *rsdA* in the wildtype is due to the higher proportion of pre-mRNA than mature mRNA.

That being said, I still feel that the revised manuscript has been much improved to merit its publication in Nature communications once the authors provide at least some plausible explanations about my questions above.

Thank you very much for your comments. According to Fig. 5, deletion of *csdA* promotes splicing of *rsdA* intron 2 and a decrease in mRNA abundance. Therefore, we hypothesized that the over-splicing of *rsdA* intron 2 in $\Delta csdA$ mutant caused the instability of newborn mRNA and affected the level of mature mRNA. As we measured in the $\Delta csdA$ mutant, the mRNA stability of *rsdA* was significantly reduced compared to the control. Therefore, the model in lines 385-386 was corrected to “we hypothesized that in the metabolic balance of the fungus when the abundance of *rsdA* expression is reduced, CsdA may promote *rsdA* expression by regulating the stability of intron 2 splicing, thereby producing normal levels of RsdA protein that interacts with CsdA to regulate fungal secondary metabolism”.

To investigate whether the reduction in *rsdA* mRNA abundance was at the transcriptional or post-transcriptional level (This is also reviewer 2's question that we answered), we separately determined the relative abundance of *rsdA* pre-mRNA (transcriptional) and mature mRNA (post-transcriptional) in $\Delta csdA$ mutant compared to control according to the designed primers. The results showed that the abundance of *rsdA* mature mRNA was significantly reduced in $\Delta csdA$ mutant compared with control, suggesting the trends to post-transcriptional level regulation. It is also possible that” *rsdA* mature mRNA has a shorter half-life than pre-mRNA resulting in lower *rsdA* expression in $\Delta csdA$ mutant” as you said.

REVIEWERS' COMMENTS

Reviewer #2 (Remarks to the Author):

The novel aspect of this manuscript is the finding that a central regulator of fungal secondary metabolism RsdA interacts with the RNA-binding protein CsdA in a complex (see title of the manuscript). Thus, we need to know at least one clear function of the complex.

Currently, the functional aspect of the complex is mainly based on a genetic analysis of deletion strains. The authors follow the logic that deletion of RsdA causes alterations in fungal secondary metabolites. Also CsdA mutants causes alterations in secondary metabolites. Both strains exhibit differences in conidiation indicating that they have import functions independent of each other. However, those secondary metabolites that are mis-regulated in both deletion strains might indicate the function of the CsdA/RsdA complex. Unfortunately, the situation is more complex, since the authors identified a very complicated regulatory network. CsdA regulates the expression of RsdA. Thus, *csdA* mutants affect RsdA and thus, we do not know which secondary metabolites are regulated by the complex and which are regulated by the altered RsdA expression in the absence of CsdA.

In order to address this one needs to dissect the regulatory network. This is not my project and the authors are responsible to develop an experimental strategy to show this convincingly. I suggested to use CsdA mutants that no longer interact with RsdA. Thus, if CsdA is not able to interact with RsdA the changes could be assigned to the complex. To address this, the authors use two mutants H26-Y29 and N-terminal deletions. In an ideal situation both mutants should show identical mis-regulation, because both can no longer interact with RsdA. However, the overlap is only 28% indicating that the simple approach might not valid. Maybe only in one of the mutants the RsdA splicing is affected or there might be other explanations.

So far we are at the level "CsdA and RsdA alone or together, regulate fungal metabolic processes ..."
Lane 182

The current state of studying the function of the complex is given here (starting from lane 223)
"to investigate the individual function of CsdA, RsdA and complex, we performed functional enrichment and metabolic analysis from transcriptomic and metabolomic levels. From the transcriptomic level, we reveal that the functions of CsdA are involved in carbon metabolism and TCA cycle, RsdA are involved in carbon metabolism and amino acid metabolism, and CsdA-RsdA complex are involved in carbon metabolism and terpenoid biosynthesis. From the metabolomic level, the metabolic ion peaks of 2830 (39%) and 3923 (55%) were significantly changed in the mutation of the binding sites H26-Y29 in the complex or complete deletion of the interaction region (N-terminus of CsdA), respectively. Among them, 1977 (28%) of the differentially regulated metabolic ion peaks were co-regulated in CsdAH26A-Y29A and CsdAΔN-terminal mutants compared with Δ*csdA* or Δ*rsdA* mutant, suggesting that this complex has an individual function in metabolite regulation (Supplementary Fig. 11)."

I do not understand how these conclusions were achieved. The data are given in Supplementary Figure 11 indicating that the data are not of main importance for the manuscript. We are still left with the question: What is the function of the complex ?

In essence, major point 1 is still valid and not addressed to merit publication in Nature Communications
Regarding Major criticism point 2:

In order to determine mRNA stability it is crucial to stop transcription, e.g. with inhibitors, and measure the reduced mRNA amount over time. Currently, the term mRNA stability can only be used with great caution in the manuscript. The authors determined the abundance of spliced and unspliced mRNAs. Using a strain expressing RsdA without intron 2 would be a possibility to dissect the function of CsdA on the expression of RsdA. However, as mentioned above this is not my project and the authors could also use alternative strategies.

Thus, my major points of criticism are still not addressed. To clarify the situation, I would like to suggest that the editor and reviewer 4 judge my criticism. If they feel that my criticism is too harsh and that the current version is of sufficient quality for Nat comm. I have no problem to be overruled.

Reviewer #4 (Remarks to the Author):

The authors have addressed my concerns as well as the other reviewers' points. The revised manuscript became much more clear and I would recommend its publication in Nature communications.

Reviewer #2 (Remarks to the Author):

The novel aspect of this manuscript is the finding that a central regulator of fungal secondary metabolism RsdA interacts with the RNA-binding protein CsdA in a complex (see title of the manuscript). Thus, we need to know at least one clear function of the complex.

Currently, the functional aspect of the complex is mainly based on a genetic analysis of deletion strains. The authors follow the logic that deletion of RsdA causes alterations in fungal secondary metabolites. Also, CsdA mutants causes alterations in secondary metabolites. Both strains exhibit differences in conidiation indicating that they have import functions independent of each other. However, those secondary metabolites that are mis-regulated in both deletion strains might indicate the function of the CsdA/RsdA complex. Unfortunately, the situation is more complex, since the authors identified a very complicated regulatory network. CsdA regulates the expression of RsdA. Thus, *csdA* mutants affect RsdA and thus, we do not know which secondary metabolites are regulated by the complex and which are regulated by the altered RsdA expression in the absence of CsdA.

In order to address this one needs to dissect the regulatory network. This is not my project and the authors are responsible to develop an experimental strategy to show this convincingly. I suggested to use CsdA mutants that no longer interact with RsdA. Thus, if CsdA is not able to interact with RsdA the changes could be assigned to the complex. To address this, the authors use two mutants H26-Y29 and N-terminal deletions. In an ideal situation both mutants should show identical mis-regulation, because both can no longer interact with RsdA. However, the overlap is only 28% indicating that the simple approach might not valid. Maybe only in one of the mutants the RsdA splicing is affected or there might be other explanations.

So far we are at the level “CsdA and RsdA alone or together, regulate fungal metabolic processes ...” Lane 182

The current state of studying the function of the complex is given here (starting from lane 223)

“ to investigate the individual function of CsdA, RsdA and complex, we performed functional enrichment and metabolic analysis from transcriptomic and metabolomic levels. From the transcriptomic level, we reveal that the functions of CsdA are involved in carbon metabolism and TCA cycle, RsdA are involved in carbon metabolism and amino acid metabolism, and CsdA-RsdA complex are involved in carbon metabolism and terpenoid biosynthesis. From the metabolomic level, the metabolic ion peaks of 2830 (39%) and 3923 (55%) were significantly changed in the mutation of the binding sites H26-Y29 in the complex or complete deletion of the interaction region (N-terminus of CsdA), respectively. Among them, 1977 (28%) of the differentially regulated metabolic ion peaks were co-regulated in CsdA^{H26A-Y29A} and CsdA^{ΔN-terminal} mutants compared with $\Delta csdA$ or $\Delta rsdA$ mutant, suggesting that this complex has an individual function in metabolite regulation (Supplementary Fig. 11).”

I do not understand how these conclusions were achieved. The data are given in Supplementary Figure 11 indicating that the data are not of main importance for the manuscript. We are still left with the question: What is the function of the complex?

In essence, major point 1 is still valid and not addressed to merit publication in Nature Communications

Regarding Major criticism point 2:

In order to determine mRNA stability it is crucial to stop transcription, e.g. with inhibitors, and measure the reduced mRNA amount over time. Currently, the term mRNA stability can only be used with great caution in the manuscript. The authors determined the abundance of spliced and unspliced mRNAs. Using a strain expressing RsdA without intron 2 would be a possibility to dissect the

function of CsdA on the expression of RsdA. However, as mentioned above this is not my project and the authors could also use alternative strategies.

Thus, my major points of criticism are still not addressed. To clarify the situation, I would like to suggest that the editor and reviewer 4 judge my criticism. If they feel that my criticism is too harsh and that the current version is of sufficient quality for Nat comm. I have no problem to be overruled.

Response:

Thanks again for your insightful comments. Your suggestions will help us to further study the mechanism of CsdA/RsdA. In our study, we have identified the interaction sites and pre-mRNA splicing sites of CsdA, RsdA, and CsdA/RsdA complexes by genetic mutation and biochemical approaches, thus characterizing their functions. Next, we will obtain the crystal structure of the CsdA/RsdA complex and conduct an in-depth study on the regulation of fungal secondary metabolism. To clarify the novelty and limitations of our study, we have reorganized a discussion section.

Reviewer #4 (Remarks to the Author):

The authors have addressed my concerns as well as the other reviewers' points. The revised manuscript became much more clear and I would recommend its publication in Nature communications.

Thanks again for your positive comments of our work.